# Evaluating Habitat Suitability for the Endangered *Sinojackia xylocarpa* (Styracaceae) in China Under Climate Change Based on Ensemble Modeling and Gap Analysis

**DOI:** 10.3390/biology14030304

**Published:** 2025-03-17

**Authors:** Chenye Hu, Hang Wu, Guangfu Zhang

**Affiliations:** Jiangsu Key Laboratory of Biodiversity and Biotechnology, School of Life Sciences, Nanjing Normal University, Wenyuan Road, Nanjing 210023, China; 09220610@njnu.edu.cn (C.H.); 09220318@njnu.edu.cn (H.W.)

**Keywords:** Biomod2, conservation, environmental factors, population centroid, suitable range

## Abstract

*Sinojackia xylocarpa*, endemic to China, has high value in landscaping. However, little is known about how it responds to climate change. We built an ensemble model in the Biomod2 package to forecast its potential distribution and evaluated its current protective effectiveness in China. The results showed that the four key influencing environmental factors were precipitation of the driest quarter, mean temperature of the warmest quarter, precipitation of the warmest quarter, and elevation. This species was mainly distributed in southeast China, with an area of 697,200 km^2^, accounting for 7.26% of China’s total territory. However, only 3.91% was located within national or provincial nature reserves. Under future climates, it would averagely decrease by 10.97% in suitable areas compared to the current, with more fragmented habitats. Therefore, our findings first demonstrate that future climate change may have an adverse effect on its distribution. We recommend conducting a supplementary investigation within the projected suitable range and establishing new conservation sites for *S. xylocarpa* in China. Moreover, this study provides a general picture for conserving other endangered *Sinojackia* species under global warming.

## 1. Introduction

Climate change has a profound effect on plant growth and distribution on a global scale [1,2]. Endangered trees are usually more susceptible to climate change in distribution relative to non-endangered ones. Take *Glyptostrobus pensilis* (Staunton ex D. Don) K. Koch as an example. This endangered tree might shrink to varying degrees in suitable habitat under different future climate scenarios [3]. More importantly, these endemic and endangered species usually have restricted distribution ranges and small population sizes. Moreover, their habitats are probably separated, thereby leading to low genetic diversities in most cases. Therefore, future climate change may pose a severe threat to such plants.

*Sinojackia* is an oligotypic genus from the Styracaceae family [4]. All species of this genus are endemic to China and endangered. Therefore, they have been protected by the Chinese government since 2021. Due to morphological variation and spontaneous hybridization between different species, there is controversy about interspecific delimitation within this genus. According to *Flora of China*, the *Sinojackia* genus comprises a total of five species and one variety [5]. Besides *Sinojackia xylocarpa* Hu, it also includes *S. rehderiana* Hu, *S. henryi* (Dümmer) Merr., *S. sarcocarpa* L. Q. Luo [6], *S. microcarpa* Tao Chen bis and G. Y. Li [7] and the variety of *S. xylocarpa*—*S. xylocarpa* var. *leshanensis* L. Q. Luo [8]. Later on, the *Sinojackia* genus also included *S. dolichocarpa* C. J. Qi, *S. huangmeiensis* J. W. Ge and X. H. Yao, and *S. oblongicarpa* Tao Chen bis and T. R. Cao. Presently, based on key taxonomic traits and related molecular phylogenetic evidence, *S. dolichocarpa* has been assigned to a new genus—*Changiostyrax* Tao Chen [9,10], *S. oblongicarpa* has been identified as the synonym of *S. sarcocarpa* [11], and *S. huangmeiensis* as the synonym of *S. xylocarpa* [12]. As a result, now there are only five species in the genus of *Sinojackia* in China.

Currently, there are few studies on the distribution prediction of the genus *Sinojackia.* Yang et al. (2020) used MaxEnt to predict the current range of this genus involving seven species in China, and they pointed out that its suitable distribution would decline in the future (2050s and 2070s) [13]. Conversely, Feng and Zhang (2024) applied MaxEnt to project the suitable distribution of *Sinojackia* comprising eight species, and they concluded that this genus would expand in the highly suitable distribution in the future (2081–2100) [14]. Such a difference is mainly due to the sampling representativeness like the number of species and their occurrence points. The former used 58 distribution points, while the latter only used 15 ones for this genus in the final model. In addition, both studies contain species that actually do not belong to the genus *Sinojackia*, such as *Changiostyrax dolichocarpus*. Indeed, genus and species are two different taxonomic categories. The potential distribution of a genus can be modeled based on its species’ occurrence records unless these species have niche conservatism, similar ecological features, and restricted geographical ranges [15,16].

However, to the best of our knowledge, there are no such studies regarding the potential distribution of any species from the *Sinojackia* genus. *S. xylocarpa* was designated as the type species of the genus as early as 1929 by a Chinese botanist Hu Hsien-Hsu [4]. According to *Flora of China*, *S. xylocarpa* is only found in Nanjing, Jiangsu Province, eastern China [17]. According to *Flora of Jiangsu*, this species is endemic to Jiangsu Province and is mainly distributed in Nanjing [18]. Its specimen collections show that *S. xylocarpa* was once wildly distributed in Mufu Mountain and Laoshan Mountain of Nanjing, and Baohua Mountain of Zhenjiang within southern Jiangsu [19]. However, because of the deterioration of the natural environment and the impact of anthropogenic activities such as quarrying and mining, there are no wild populations of *S. xylocarpa* at Mufu Mountain, Nanjing [6]. This species is even considered extinct in the wild in China [20,21]. In recent years, with the extensive investigation of its natural populations, *S. xylocarpa* has been discovered in the provinces of Anhui and Zhejiang, eastern China, and Hunan Province, central China [12,22]. For example, a wild population of *S. xylocarpa*, with more than 200 individuals, was found in 2023 in Majiazui, Yiyang City, Hunan Province (https://lyj.hunan.gov.cn/lyj/xxgk_71167/gzdt/xlkb/xsqxx/202304/t20230418_29316935.html, last accessed on 5 December 2024). In 2024, 11 bushes with *S. xylocarpa* saplings were found in Huangli Mountain, Chaohu City, Anhui Province (http://www.ahwang.cn/hefei/20241006/2754153.html, last accessed on 5 December 2024). Meanwhile, according to our field survey in the past two years, its wild populations occurred in various sites such as Laoshan, Nanjing City, Jiangsu Province, and Wuwei, Wuhu City, Anhui Province (Figure 1c,d). Additionally, due to the taxonomic revision, *S. huangmeiensis* has been merged into *S. xylocarpa*. In summary, we believe that the distribution of *S. xylocarpa* in China is geographically restricted and discontinuous, but its actual distribution range is still unclear.

Species-distribution models (SDMs) link species presence, absence, or abundance information with environmental variables to predict its potential locations and quantities [23]. At present, it has been widely used in multiple fields like conservation biology, ecological invasion, and habitat suitability assessment [24,25]. For example, SDMs were employed to predict climatically suitable habitats of the endemic and endangered *Parrotia subaequalis* (H. T. Chang), R. M. Hao, and H. T. Wei in China [26]. More recently, it has been noted that an ensemble model comprising multiple individual models can improve the accuracy of model predictions relative to a single model [27]. Biomod2, including ten species-distribution models, is a program package developed by Wilfried Thuiller et al. for SDM applications [28]. Presently, it is widely applied in potential distribution prediction for endangered species [29].

In this study, we first collected data on the distribution points of *S. xylocarpa* and related environmental variables, then used Biomod2 to screen suitable models to generate an ensemble model, and finally used the ensemble model to predict its potential distribution in China. Specifically, we focused on the following issues. (1) We identified key environmental factors affecting the distribution of *S. xylocarpa*; (2) We projected its potential distribution areas under different climate scenarios in the past, current, and future and determined its centroid shift; (3) In addition, we further assessed its conservation status by overlaying the resulting suitable habitats with existing nature reserve layers in China. The purpose of this study is to provide a scientific basis for conservation recommendations for endangered *S. xylocarpa*, as well as a conservation reference for other endangered *Sinojackia* species in China.

## 2. Materials and Methods

### 2.1. Acquisition of Geographic Distribution Data

*S. xylocarpa* is a light-demanding tree and grows well in a warm, humid climate. Moreover, it places little emphasis on soil [30]. As a native tree, *S. xylocarpa* has high ornamental value [6]. In spring, it produces small white delicate flowers (Figure 1a), and in autumn, it bears many conical fruits with long slender pedicels, like a balanced weight set (“Chengtuo” in Chinese) hanging in the tree (Figure 1b). Recent studies have shown that *S. xylocarpa* has a low germination rate in the wild, resulting from its physiological seed dormancy, hard seed coat, and high microspore abortion rate in floral organogenesis [31]. Coupled with external factors such as climate change and habitat destruction [6,32,33], *S. xylocarpa* is on the brink of extinction in China. Therefore, this species was listed as a national secondary protected plant species in 1999. In addition, it has been listed as one of the key protected wild plants of China since 2021 (https://www.forestry.gov.cn/, last accessed on 27 September 2024). In addition, it has been ranked as a “Vulnerable” (VU) species in the IUCN Red List (https://www.iucnredlist.org/, last accessed on 27 September 2024).

The data regarding the wild distribution of *S. xylocarpa* were obtained through several sources. (1) Investigating in the field: From the spring of 2022 to the autumn of 2024, we surveyed the wild population of *S. xylocarpa* in Anhui (i.e., Hefei, Wuhu, Xuancheng), Jiangsu (i.e., Changzhou, Nanjing, Zhenjiang), Zhejiang (i.e., Hangzhou, Huzhou, Ningbo), and other provinces in eastern China to determine its distribution. Simultaneously, we located the *S. xylocarpa* populations with GPS and recorded their geographical coordinates (i.e., latitude and longitude). (2) Browsing related websites: We accessed botanical websites, including the Plant Photo Bank of China (PPBC, http://ppbc.iplant.cn/, last accessed on 5 December 2024), the Chinese Virtual Herbarium (CVH, https://www.cvh.ac.cn/, last accessed on 5 December 2024), and the National Specimen Information Infrastructure (NSII, http://nsii.org.cn/, last accessed on 5 December 2024). (3) Consulting published literature and related reports: We examined *Flora of China*, provincial floras, and related checklists that listed the specific name, synonym, and Latin name of *S. xylocarpa*. Furthermore, we conducted searches for published literature and pertinent articles [12,34]. Accordingly, we obtained a total of 104 distribution points for *S. xylocarpa.* After removing error duplicate points, we collected 22 natural distribution records of this species.

Following preliminary data collation, we employed Spatially Rarefy Occurrence Data for Species-Distribution Models (SDMs) in the SDMs toolbox (version 2.6), ensuring a single occurrence point per 1 km × 1 km grid [35]. Such a filtering approach is particularly useful for species with limited occurrence points, as it maximizes the number of spatially independent localities [36]. Furthermore, finer-scale data are considered to better reflect climatic conditions experienced by species [37]. Finally, we obtained latitude and longitude data for 21 points of *S. xylocarpa* (Figure 2; Appendix A).

### 2.2. Selection and Filtering of Environmental Variables

The environmental data selected for this study are categorized into three groups: climate, terrain, and soil [38,39,40]. They encompass the climatically historical periods (the Last Interglacial period, approximately 12,000–14,000 years ago, and the Middle Holocene, around 5000–7000 years ago), the current, and the future periods (2050s: 2041–2060; 2070s: 2061–2080) [29,41]. Given the considerable temporal separation from the present epoch and the profound transformations Earth’s environment has undergone, only bioclimatic variables from two distinct paleoclimate periods are selected for prediction. In the current and future periods, topographic and soil variables will be used, besides bioclimatic variables. We downloaded 19 bioclimatic factors for the three periods from WorldClim (https://www.worldclim.org/, last accessed on 5 December 2024). Then, we standardized the resolution to 30 s (1 km × 1 km) to ensure accuracy during modeling. Considering inter-model variability and projection uncertainty, we employed an ensemble modeling approach, incorporating multiple climate models from the Coupled Model Intercomparison Project Phase 6 (CMIP6). Future bioclimatic data were derived from three global climate models: BCC-CSM1-1 (the Beijing Climate Center Climate System Model version 1.1), CCSM4 (the Community Climate System Model version 4), and MIROC-ESM (an Earth System Model based on the model for interdisciplinary research on climate) [42]. Additionally, three Representative Concentration Pathways (RCPs) were selected to represent different greenhouse gas emission trajectories: RCP2.6 (representing a moderate emission scenario), RCP4.5 (a medium and stable emission scenario), and RCP8.5 (a high emission scenario).

Topographic data includes elevation and slope. Since these variables remain essentially unchanged over time, they are added to the models as constant variables [43]. Digital elevation data were obtained from WorldClim, and slope data were downloaded from the National Earth System Science Data Center (https://www.geodata.cn/main/, last accessed on 5 December 2024). Soil characteristics can affect the physiological growth of plants. Species-distribution models with soil data perform significantly better than those without soil information [44]. The China soil dataset (version 1.2) was downloaded from the National Qinghai–Tibet Plateau Scientific Data Center (http://www.tpdc.ac.cn/zh-hans/, last accessed on 5 December 2024), and 16 types of surface soil data were selected from the website for subsequent research.

The three types of environmental data were standardized using the WGS1984 coordinate system, and the “Extract by Mask and Clip” tool in ArcGIS 10.8 was employed to ensure that the data were confined to China. The data resolution was subsequently adjusted to the 30 s level using resampling tools. Concurrently, Pearson correlation analysis was conducted to mitigate collinearity among related environmental variables, ensuring that redundant information did not compromise the model’s predictions [45]. Environmental variables with a low contribution rate and |*r*| ≥ 0.8 were excluded from further analysis. Ultimately, we retained 10 bioclimatic variables for the Last Interglacial, 11 bioclimatic variables for the Middle Holocene, and 22 environmental variables for the current and future periods for subsequent modeling (Table 1).

### 2.3. Modeling Process

We used the Biomod2 package to generate an ensemble model to simulate the distribution range of *S. xylocarpa*. First, we combined 22 environmental variables with 21 distribution sites of *S. xylocarpa* to evaluate the 10 models, respectively, in the Biomod2 package. Then, we obtained the AUC (Area Under the Curve) and TSS (True Skill Statistic) values for each model. Since the AUC value usually ranges from 0 to 1, the closer the value is to 1, the higher the precision. It is classified as failure (0.5–0.6), poor (0.6–0.7), fair (0.7–0.8), good (0.8–0.9), and excellent (0.9–1.0) [46]. The TSS value varies between −1 and +1. The value close to 1 indicates good performance, while the value close to or below 0 indicates poor performance. It can be divided into five groups: excellent (TSS > 0.8), good (0.6–0.8), fair (0.4–0.6), poor (0.2–0.4), and fail (TSS < 0.2) [29].

At present, there is no consensus on the threshold values of AUC and TSS when evaluating ensemble model performance, so herein we followed the method of ensemble model evaluation for *Emmenopterys henryi* Oliv., an endangered tree endemic to China [27]. We selected the models with an AUC exceeding 0.8 and a TSS exceeding 0.7 from the 10 models to construct an ensemble model. Consequently, the ensemble model was composed of six individual models: generalized boosted models (GBM), flexible discriminant analysis (FDA), random forest (RF), Maximum entropy model (MaxEnt), artificial neural networks (ANN), and classification tree analysis (CTA). During the modeling process, R4.3.3 randomly generated 1000 pseudo-absence points [47,48]. Simultaneously, we randomly assigned 75% of the occurrence records to the training set and the remaining 25% to the testing set. To ensure the accuracy of the predictive model, we conducted the computation ten times and adopted the average value as the final modeling result.

### 2.4. Suitable Habitat Partitions and Centroid Shift

The results generated by the ensemble model were imported into ArcGIS 10.8 for visualization. Most species modeling techniques produce continuous suitability predictions. However, many practical applications require secondary outputs which need the establishment of thresholds. We employed the maximum sum of specificity and sensitivity (maxSSS) to determine the threshold, which is a promising approach when only limited data are available [49]. Considering that *S. xylocarpa* is an endemic and endangered species, we followed the method of Liu et al. (2013) [50]. According to the maxSSS threshold (0.1916), the potential distribution of *S. xylocarpa* was categorized into unsuitable (0.0–0.19), low-suitable (0.19–0.46), moderately suitable (0.46–0.73), and highly suitable (0.73–1.00) areas. In this study, we considered both the moderately and highly suitable areas as the suitable habitat for *S. xylocarpa* [46]. Subsequently, we calculated the suitable area for each category.

Centroid shifts can provide insights into how species change in distribution under different climate scenarios. We employed the SDMtoolbox in ArcGIS 10.8 to simulate the centroid shift of *S. xylocarpa* across various climate scenarios, encompassing both the direction and distance of movement across different periods.

### 2.5. Conservation Gap Analysis

First, we created a map layer of China’s nature reserves, excluding marine protected areas. This layer includes 464 national nature reserves and 806 provincial nature reserves [51]. The total area of the protected area was 971,800 km^2^, which was about 10.12% of China’s land area. The data on nature reserves were obtained from the World Database on Protected Areas (http://www.protectedplanet.net/, last accessed on 7 December 2024) and the Ministry of Ecological Environment of China (http://www.mee.gov.cn, last accessed on 7 December 2024). Next, we used ArcGIS 10.8 to overlay the raster model of the predicted distribution of *S. xylocarpa* in the current climate with the surface layer of the protected area to obtain the coverage of the nature reserve in the *S. xylocarpa*’s suitable habitat area and to assess the conservation gaps. The *S. xylocarpa* population in a suitable area grid is considered to be protected if the grid falls within a nature preserve [29]. Finally, we calculated the extent and proportion of its suitable areas in national nature reserves, provincial nature reserves, and these two types of reserves.

## 3. Results

### 3.1. Model Performance

We used Biomod2 to establish 10 individual models for *S. xylocarpa*. Following the selection criteria described in the Materials and Methods section, we included models with AUC > 0.8 and TSS > 0.7 in the ensemble model (Table 2).

Therefore, we selected the six models to establish an ensemble model. Except for ANN and CTA, the AUC values of the other four models were all greater than 0.9, indicating that these models reached an excellent level. Meanwhile, the TSS values of these six models were all greater than 0.7, indicating that they all had high credibility and accuracy. For example, the AUC value of MaxEnt was 0.9690, and the TSS value of CTA was 0.7864. In contrast, the AUC and TSS values of the ensemble model were 0.9960 and 0.9500, respectively, both of which were higher than those of the six individual models. Therefore, the ensemble model’s superior performance likely results from combining predictions from multiple algorithms, which reduces individual model biases and enhances predictive accuracy.

### 3.2. Main Environmental Factors

We used the ensemble model to determine the contribution rate of each environmental variable in different periods (Table 1). Among these variables affecting the distribution of *S. xylocarpa* at present, Bio17 (precipitation of driest quarter) was the highest, followed by Bio10 (mean temperature of warmest quarter), Bio18 (precipitation of warmest quarter), and then elevation. Their contribution rates were 61.0%, 9.6%, 8.9%, and 5.4%, respectively, with a cumulative contribution rate reaching 84.9%. Therefore, the top four were identified as the key environmental factors.

During the Last Interglacial period, the key environmental factors were Bio10 (mean temperature of warmest quarter) (21.7%), Bio17 (19.6%), and Bio4 (temperature seasonality) (19.1%). In the Middle Holocene period, the key factors were Bio15 (precipitation seasonality) (27.5%), Bio17 (23.0%), and Bio11 (mean temperature of coldest quarter) (15.5%), respectively. Notably, Bio17, which is the dominant factor in the current period, ranked second during both the Last Interglacial and Middle Holocene periods, suggesting a shift in climatic drivers of habitat suitability over time. Hence, the contribution of environmental factors varied across different periods, highlighting changes in habitat suitability over time.

When the presence probability was greater than 0.46, the corresponding areas were considered to be moderately or highly suitable, and we thought that they were conducive to the growth of *S. xylocarpa*. Response curves represented the relationship between environmental variables and species presence probability, reflecting the species’ biological tolerance and habitat preferences. In other words, response curves indicate ecological thresholds, where the probability of species presence changes non-linearly in response to key environmental factors.

When the precipitation of the driest quarter was greater than 90 mm, it was suitable for the survival of *S. xylocarpa.* As shown in Figure 3a, the probability of *S. xylocarpa* presence increased sharply with precipitation of the driest quarter, stabilized, and then decreased. When the mean temperature of the warmest quarter was greater than 24.6 °C, it was suitable for the survival of *S. xylocarpa*. As the mean temperature of the warmest quarter increased, the existence probability of *S. xylocarpa* first increased and then remained unchanged (Figure 3b). The suitable precipitation range for the warmest quarter was 417–763 mm. As the precipitation of the warmest quarter increased, the existence probability of *S. xylocarpa* first increased considerably and then decreased sharply (Figure 3c). *S. xylocarpa* was suitable for growth when the elevation was less than 392 m. As the elevation increased, the existence probability of *S. xylocarpa* first remained unchanged and then decreased sharply (Figure 3d). Notably, when the elevation exceeded 1000 m, its survival probability decreased to less than 0.3. As a result, the response curves of Bio17 (in Figure 3a) and Bio18 (in Figure 3c) present a unimodal pattern, while the response curves of Bio10 (in Figure 3b) and elevation (in Figure 3d) present an asymptotic pattern.

### 3.3. Potential Suitable Habitats in the Current

At present, the suitable areas for *S. xylocarpa* are mainly concentrated in southern Anhui, northern Guangxi, eastern Hubei, Hunan, southern Jiangsu, northern Jiangxi, eastern Taiwan, and northern Zhejiang (Figure 4). Some suitable areas were also predicted to be scattered in northern Fujian, southern Guangdong, eastern Guizhou, and other parts of China. Furthermore, its suitable areas were relatively more fragmented than its low ones in the current period (Figure 4). The total suitable area for *S. xylocarpa* was 697,200 km^2^, accounting for only 7.26% of China’s total land area, and the highly suitable area was 341,500 km^2^, accounting for 3.56% (Table 3). Collectively, this species was mainly distributed in the southeast of China, which is largely consistent with the surveyed distribution points.

Under the current climate, the suitable habitat of *S. xylocarpa* within the boundaries of national nature reserves was 12,800 km^2^, accounting for 1.84%. The suitable habitat of *S. xylocarpa* within the boundaries of provincial nature reserves was 15,000 km^2^, accounting for 2.15%. The coverage ratio of national and provincial nature reserves in the suitable area of *S. xylocarpa* was only 3.91%. Therefore, the vast majority of the suitable areas for *S. xylocarpa* were not effectively protected (Figure 4).

### 3.4. Potential Suitable Habitats in the Past

During the Last Interglacial, the suitable habitat for *S. xylocarpa* in China was predominantly found in southern Anhui, southern Hubei, Hunan, southern Jiangsu, northern Jiangxi, and northern Zhejiang (Figure 5a). The species exhibited a more continuous distribution pattern compared to its current fragmented range. The total suitable area amounted to 638,400 km^2^, showing a decrease of 8.43% from current levels (Table 3).

During the Middle Holocene, the suitable habitat for *S. xylocarpa* shifted slightly, mainly concentrating in southern Anhui, northern Fujian, eastern Hubei, Hunan, southern Jiangsu, northern Jiangxi, northern Taiwan and Zhejiang (Figure 5b). Compared with the Last Interglacial, the suitable habitat during the Middle Holocene presented a more fragmented pattern. The total suitable area was estimated at 645,000 km^2^, indicating a decrease of 7.49% relative to the current (Table 3).

In a word, the results show a continuous contraction and fragmentation of suitable habitats from the Last Interglacial to the Middle Holocene, indicating increasing environmental constraints for *S. xylocarpa* over time. For instance, in the Middle Holocene, increased fragmentation suggests reduced connectivity between habitats, which could have impacted species dispersal and survival.

### 3.5. Potential Suitable Habitats in the Future

Future projections indicated that the potential suitable habitat for *S. xylocarpa* primarily occurred in southern Anhui, southern Hubei, Hunan, southern Jiangsu, northern Jiangxi, northern Taiwan, and Zhejiang. However, model projections revealed varying degrees of suitable habitat contraction across these regions under most future scenarios (Figure 6).

Under six future climate scenarios, the predicted suitable area was, on average, 620,700 km^2^, which decreased by 10.97% compared to the current. Except for an increase under RCP 8.5 in the 2070s, the suitable area decreased in the other five scenarios. It was expected that under RCP 8.5 in the 2050s, the suitable area would decrease the most, which was reduced by 16.32% compared to the current. In contrast, under RCP 4.5 in the 2050s, the suitable area was expected to decrease the least, which decreased by 11.20% compared to the current situation. In addition, the highly suitable area was expected to increase in some scenarios and decrease in others. Overall, the average highly suitable area under the six future scenarios was expected to be 362,600 km^2^, which increased by 6.18% compared to the current condition. However, the average moderately suitable area under future climate was expected to be 258,100 km^2^, which decreased by 27.44% compared to the current condition. In addition, it was expected that this species would decrease in suitable areas much more in the 2050s than in the 2070s (Table 3).

Overall, the suitable area for *S. xylocarpa* under future climate scenarios, with more habitat fragmentation, was mostly smaller than under the current condition. This indicated that future climate might be unfavorable for the survival of *S. xylocarpa*.

### 3.6. Centroid Shift Under Different Climate Scenarios

The current centroid of *S. xylocarpa* was located at 114.446° E, 27.793° N. From the Last Interglacial (113.216° E, 29.201° N) to the Middle Holocene (112.682° E, 28.955° N) and then to the present, the centroid first shifted 58.70 km toward the southwest and then 215.55 km toward the southeast. Under RCP 2.6, it was expected that in the 2050s, the centroid would shift 147.91 km in the northwest direction to 114.368° E, 29.126° N, and by the 2070s, it would shift 134.57 km in the north direction to 114.446° E, 29.007° N. Under RCP 4.5, it was expected that in the 2050s, the centroid would migrate 52.96 km in the northeast direction to 114.555° E, 28.261° N, and by the 2070s, it would shift 153.22 km in the northeast direction to 115.178° E, 29.015° N. Under RCP 8.5, it was expected that in the 2050s, the centroid would shift 201.67 km in the northeast direction to 115.722° E, 29.221° N, and by the 2070s, it would shift 413.29 km in the northwest direction to 111.412° E, 30.402° N (Figure 7).

Overall, the centroid exhibited a sinuous changing pattern. The centroid of *S. xylocarpa*’s suitable habitat shifted southwest from the Last Interglacial to the Middle Holocene and then southeast to the present. In future projections, the centroid generally moves northeast under all RCP scenarios, indicating a shift toward higher latitudes as the climate warms. This northeastward shift reflects a common response of species to climate change, moving toward cooler regions as temperatures rise. However, such shifts may result in habitat loss if suitable areas become fragmented or unavailable.

## 4. Discussion

### 4.1. Model Selection and Evaluation

It is generally believed that ensemble models make better predictions of species distribution than single models [27]. Our ensemble model results confirmed this observation. In this study, we forecast the potential distribution of the endangered tree species *S. xylocarpa* with each of the ten individual models in the Biomod2 platform separately. The results showed that there were six models with the value of AUC > 0.8 and TSS > 0.7. Next, we combined the six models into an ensemble model, whose AUC and TSS were all above 0.9 (Table 2). This indicated that such an integrated model outperformed individual models. Subsequently, we used this model to predict the current distribution of *S. xylocarpa* and noticed that the prediction was generally consistent with the known distribution points. This indicated that the ensemble model had demonstrated good predictive accuracy. Thus, we employed this model to project the suitable distribution of *S. xylocarpa* under different climate scenarios of past, present, and future.

### 4.2. Key Influencing Factors of S. xylocarpa

The model projections show that the top three factors affecting the current potential distribution of *S. xylocarpa* are Bio17, Bio10, and Bio18. The sum of their contribution rates is nearly 80%, which suggests that the main factors limiting the distribution of *S. xylocarpa* are bioclimates rather than topography or soil. Among these climatic factors, Bio17 has the largest contribution rate, exceeding 60%, which is much larger than Bio10 (9.6%) and Bio18 (8.9%). This indicates that precipitation-related variables may play a greater role in shaping the distribution of *S. xylocarpa* than temperature-related variables. Unlike Zhu et al. (2024), who found temperature-related variables to be dominant, our results emphasize the importance of precipitation, likely due to the larger sample size (21 distribution points vs. 2) [31]. Our results conform with the tree traits that this species prefers to grow in warm and moist conditions [30].

Our results also show that the contribution rate of elevation is 5.4% among 22 variables, which ranks fourth. This indicates that besides climatic factors, elevation is also one of the important factors limiting the distribution of *S. xylocarpa*. Yang et al. (2018) used a self-organizing map (SOM) to analyze the wild *S. xylocarpa* community in Laoshan mountain of Nanjing, Jiangsu Province, eastern China, and found that elevation was the main factor affecting the growth and distribution of *S. xylocarpa* [52]. This is roughly in accord with our results.

Therefore, our study suggests that *S. xylocarpa* prefers to grow in low-altitude areas with a warm and humid climate, which conforms with the phenomenon that most of the known *S. xylocarpa* populations concentrate in the subtropical hilly areas of southeastern China (Field observation by corresponding author).

### 4.3. Current Suitable Area of S. xylocarpa

The resulting model outcomes show that the current suitable area for *S. xylocarpa* is 697,200 km^2^, accounting for only 7.26% of China’s total land area. It is mainly distributed in Anhui, Guangxi, Hubei, Hunan, Jiangsu, Jiangxi, Taiwan, and Zhejiang in China (Figure 4). Given that the fruit of *S. xylocarpa* are drupes (Figure 1b) with a hundred-kernel weight of 98.4 g [53], it seems unlikely for its seeds to disperse from the Chinese mainland to Taiwan because Taiwan Strait separates them with a minimum width of 130 km.

Actually, up to now, there is no record of its wild populations in Taiwan Province [54]. This indicates that the actual geographical range of an endangered tree species depends largely on its suitable habitat, as well as on its origin and evolution, on propagule dispersion, and interaction with different species [25]. Therefore, we think that the suitable distribution area of *S. xylocarpa* covers seven provinces except Taiwan in China. However, according to *Flora of China, S. xylocarpa* occurs only in Jiangsu Province. According to the newly published monograph *National Key Protected Wild Plants of China* [55], it is distributed in Nanjing in Jiangsu Province, Hangzhou in Zhejiang Province, Shanghai, and Wuhan in Hubei Province, which differs from our results. In fact, as early as 1987, *Flora of China* recorded that *S. xylocarpa* occurred in Nanjing and that it was cultivated in large cities like Hangzhou, Shanghai, and Wuhan [17]. Therefore, it is probably not true about its distribution description by Jin et al. (2023) [55]. More recently, a wild population has been found in the Cixi Mountain area of Ningbo, Zhejiang Province [31]. We also found its wild community in the Wuwei hilly area of Wuhu, Anhui Province, in 2024 (Figure 1d). Therefore, our results indicate that *S. xylocarpa* has a much larger suitable habitat in China than it is known.

### 4.4. The Change in Suitable Areas in the Past and Future

According to modeling analysis, *S. xylocarpa* had a suitable area of 638,400 km^2^ in the Last Interglacial (LIG) and moderately expanded to 645,000 km^2^ in the Middle Holocene (MH). Compared with the current climate scenario, their suitable areas decreased by 8.43% and 7.49%, respectively. Furthermore, habitat fragmentation of *S. xylocarpa* was increasing from LIG to MH (Figure 4). Overall, there was an increasing trend in suitable habitats for *S. xylocarpa* over time. This is likely due to global warming since the Holocene, with higher temperatures and more precipitation [56], which favors the expansion of *S. xylocarpa* populations. Furthermore, this may also be related to the bottleneck effect of *S. xylocarpa* populations after experiencing multiple glacial periods [31].

Under future climate scenarios, excluding RCP 8.5 in the 2070s, the suitable areas for *S. xylocarpa* will decrease, ranging from 11.2% to 15.06%, in the five remaining scenarios. Overall, the suitable habitat for *S. xylocarpa* is expected to be reduced by an average of 10.97% under future scenarios compared to the current (Table 3). Moreover, its habitat fragmentation will be exacerbated under future scenarios compared to the current (Figure 5). This suggests that future climate change may be unfavorable for the growth and distribution of *S. xylocarpa*.

In addition, the shift direction of *S. xylocarpa* in the future is largely to the north, especially to the northeast (Figure 6), which is similar to these endangered tree species like *Taxus wallichiana* var. *Mairei*, *T. wallichiana* var. *chinensis* [57] and *Emmenopterys henryi* [27]. This may be related to the tree traits. Just as stated above, the most critical factor influencing the current distribution of *S. xylocarpa* is Bio17 (precipitation of driest quarter), followed by Bio10 (mean temperature of warmest quarter) (Table 2). Specifically, *S. xylocarpa* prefers to grow in relatively warm and humid habitats in China.

### 4.5. Conservation Implications for S. xylocarpa

*S. xylocarpa* has a suitable area of 697,200 km^2^, spanning seven provinces, and is mainly distributed in the low-elevation areas of southeastern China. This is quite different from previous views. For example, it is usually assumed that its wild populations only occur in Jiangsu Province, eastern China [18,58], i.e., this species has long been believed to be endemic to Jiangsu. Given that *S. xylocarpa* usually forms patchy populations with small deme sizes, it is recommended that supplementary surveys be carried out in suitable areas for *S. xylocarpa*, especially in highly suitable areas (Figure 4), such as eastern Hunan, northern Jiangxi, and northern Zhejiang in the future.

In addition, Zhu et al. (2024) pointed out that, like other endangered tree species, *S. xylocarpa* had a low genome-wide nucleotide diversity [31]. However, their study only sampled two locations: Nanjing in Jiangsu Province and Ningbo in Zhejiang Province. Our results indicate that *S. xylocarpa* is distributed across multiple provinces, and its different populations are often isolated from each other (Figure 4). Therefore, it is advised that more samples should be collected from various locations to reveal the genetic diversity and genetic structure of *S. xylocarpa*.

Our analysis of overlaying the suitable distribution of *S. xylocarpa* with national and provincial nature reserves reveals that only 1.84% of its suitable area falls within national nature reserves and 2.15% within provincial nature reserves. This highlights that over 90% of its suitable habitat is in a zero-protection status. According to the IUCN Red List, this species is ranked as vulnerable. However, such an assessment is based on the data from 1998 (www.iucnredlist.org/species/32374/9701730, accessed on 3 February 2025). Indeed, it belonged to the “endangered” category in China [59] and “endangered” in Jiangsu Province [60]. Its endangerment can be ascribed to the following reasons. The species is generally distributed in low-altitude areas of southeastern China, in which there are usually intense human activities, such as logging, grazing, road-building, and touring [61]. This inevitably results in habitat fragmentation or habitat destruction. Recent research has confirmed that highly lignified and fibrotic pericarps inhibit the seed germination of *S. xylocarpa* [31]. Other researchers hold that its compact endosperm is also a mechanical barrier to embryo germination [53]. Additionally, it is noted that its seeds usually need to be treated with low temperatures in winter after seed maturation in the field. Afterward, its hard seed coat will be decayed and its seeds can germinate in the next spring [53]. Our results also show that the current distribution areas of *S. xylocarpa* are isolated from each other, with severe habitat fragmentation. Therefore, we propose conservation strategies for *S. xylocarpa,* such as *in-situ* conservation, *exsitu* conservation, and restoration. *In-situ* conservation involves expanding existing nature reserves in highly suitable areas, and *ex-situ* conservation focuses on developing seed banks and botanical gardens to maintain genetic diversity. Additionally, restoration strategies for this species highlight reconnecting fragmented habitats through ecological corridors.

Furthermore, *Sinojackia* is a monophyletic genus, and currently consists of five species in China. The vast majority of *Sinojackia* species have small population sizes and narrow distribution areas, and all species are endemic to China. Now, they are all listed as national secondary protected wild plants [55]. This study, taking *S. xylocarpa* as a representative species, for the first time employs an ensemble model to determine its suitable distribution, identifies the key factors influencing its distribution, and predicts the impact of climate change on its geographical distribution across different periods. Such a study can provide a valuable reference for the conservation of other endangered *Sinojackia* species in the future. In addition, our study also highlights that for a taxon with restricted or endemic distribution at the local scale, it is more appropriate to forecast the habitat suitability at the level of species than at the level of genus.

## 5. Conclusions

Here, we developed an ensemble model consisting of six models to project the potential distribution of the endangered tree *S. xylocarpa* endemic to China across different climate scenarios. The outcomes indicate that climate change may have an adverse effect on its suitable area and habitat integrity. This study is the first to demonstrate that this species is mainly distributed in southeast China, with a suitable area of 697,200 km^2^, which is larger than known. Nevertheless, more than 90% of the suitable areas are outside national or provincial nature reserves in China. Therefore, our study contributes to the conservation, management, and cultivation of *S. xylocarpa* and can also provide useful information for other endangered *Sinojackia* species in China.

## Figures and Tables

**Figure 1 biology-14-00304-f001:**
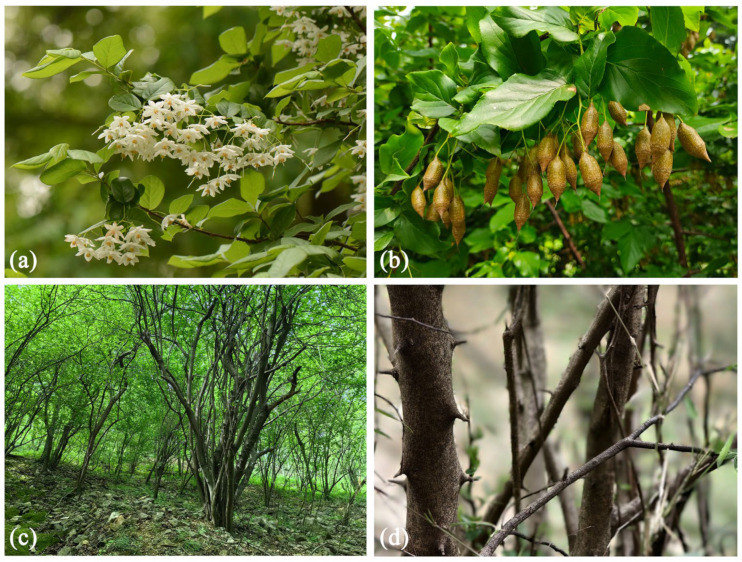
Photos of endangered *Sinojackia xylocarpa* in the field. (**a**) Individuals in Nanjing, Jiangsu Province; (**b**) Individuals in Wuwei, Anhui Province; (**c**) Blooming flowers; (**d**) Ovoid fruit (drupes). The photos were taken by Guangfu Zhang.

**Figure 2 biology-14-00304-f002:**
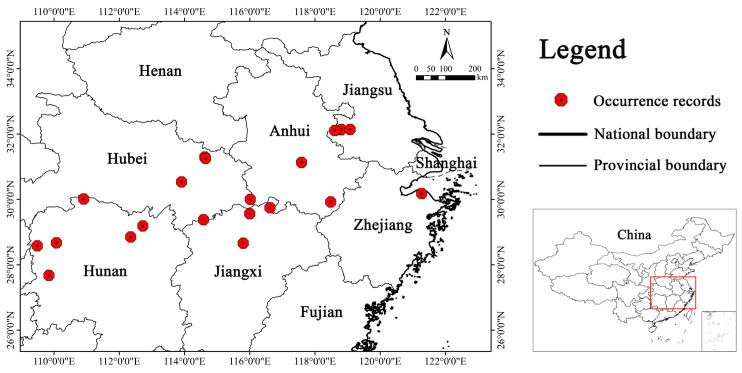
Distribution records of endangered *S. xylocarpa* in China.

**Figure 3 biology-14-00304-f003:**
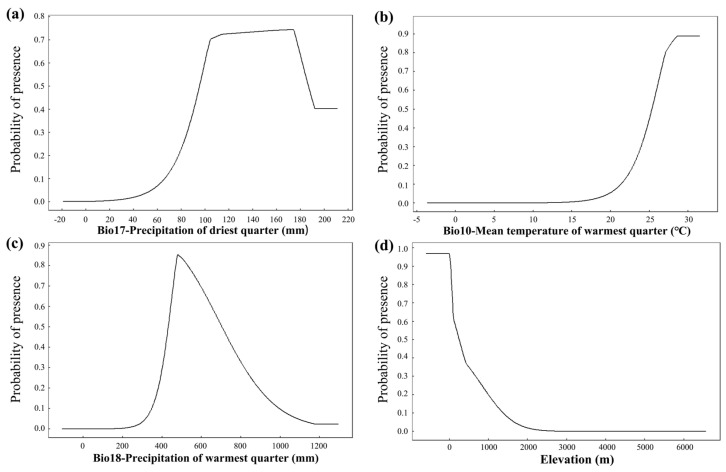
Response curves of *S. xylocarpa* to crucial environmental variables. (**a**) Precipitation of driest quarter (Bio17, mm); (**b**) Mean temperature of warmest quarter (Bio10, °C); (**c**) Precipitation of warmest quarter (Bio18, mm); (**d**) Elevation (m).

**Figure 4 biology-14-00304-f004:**
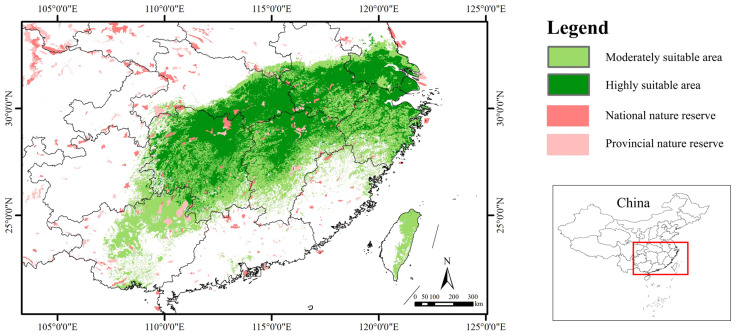
The current suitable habitat of *S. xylocarpa* overlaps with national and provincial nature reserves in China.

**Figure 5 biology-14-00304-f005:**
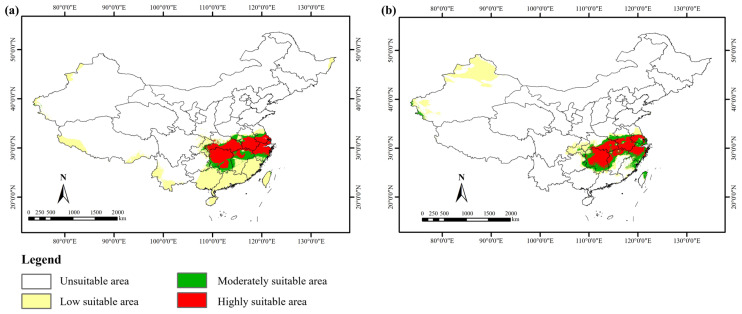
Spatial distribution change in suitable habitats under different past scenarios. (**a**) Last Interglacial (LIG); (**b**) Middle Holocene (MH).

**Figure 6 biology-14-00304-f006:**
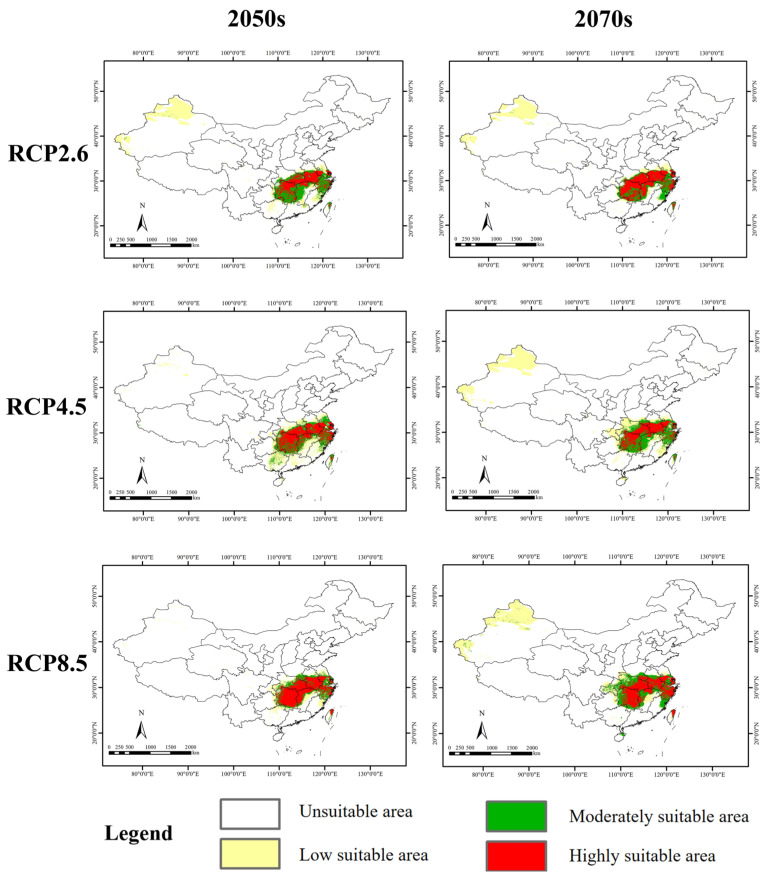
Spatial distribution change in suitable habitats under six future scenarios.

**Figure 7 biology-14-00304-f007:**
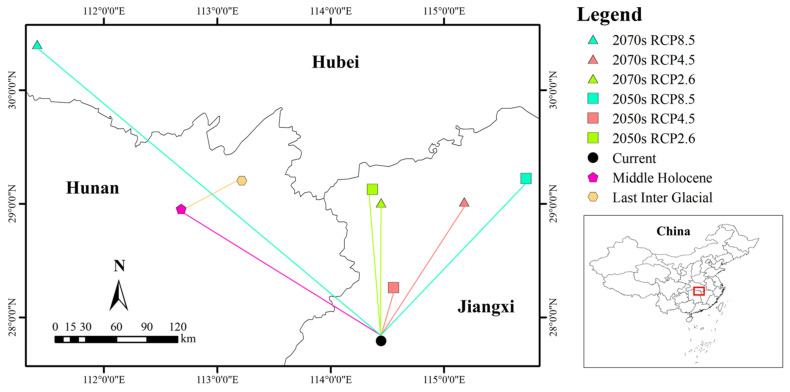
Centroid shift of suitable habitats for *S. xylocarpa* in China under different climate scenarios.

**Table 1 biology-14-00304-t001:** Environmental variables and percent contribution of variables (in bold font) used in the final ensemble model across different climate scenarios. Note: LIG and MH mean the Last Interglacial and the Middle Holocene, respectively.

Category	Variable	Description	Unit	Percent Contribution (%)		
				LIG	MH	Current
Bioclimate	Bio1	Annual mean temperature	°C			
	Bio2	Mean diurnal range (mean of monthly (max temp–min temp))	°C	5.1	0.5	
	Bio3	Isothermality ((Bio2/Bio7) × 100)	%	0.4	2.8	2.2
	Bio4	Temperature seasonality (standard deviation × 100)	-	**19.1**		1.5
	Bio5	Max temperature of warmest month	°C		1.2	
	Bio6	Min temperature of coldest month	°C			0.6
	Bio7	Temperature annual range (Bio5–Bio6)	°C		2.1	
	Bio8	Mean temperature of wettest quarter	°C	3.1	7.3	
	Bio9	Mean temperature of driest quarter	°C		1.9	2.7
	Bio10	Mean temperature of warmest quarter	°C	**21.7**		**9.6**
	Bio11	Mean temperature of coldest quarter	°C	**16.6**	**15.5**	
	Bio12	Annual precipitation	mm			4.4
	Bio13	Precipitation of wettest month	mm	3.7	8.7	
	Bio14	Precipitation of driest month	mm			
	Bio15	Precipitation seasonality (coefficient of variation)	-	6.5	**27.5**	0.9
	Bio16	Precipitation of wettest quarter	mm			
	Bio17	Precipitation of driest quarter	mm	**19.6**	**23.0**	**61.0**
	Bio18	Precipitation of warmest quarter	mm	4.2	**9.5**	**8.9**
	Bio19	Precipitation of coldest quarter	mm			
Topography	Elevation	-	m			**5.4**
	Slope	-	°			0.2
Soil	T-BS	Topsoil Base Saturation	%			0.5
	T-CaCO_3_	Topsoil Calcium Carbonate	%			0.1
	T-CEC-CLAY	Topsoil CEC (clay)	-			0.7
	T-CEC-SOIL	Topsoil CEC (soil)	-			0.1
	T-CLAY	Topsoil Clay Fraction	%			
	T-ECE	Topsoil Salinity (Elco)	S/m			
	T-ESP	Topsoil Sodicity (ESP)	-			0.1
	T-GRAVEL	Topsoil Gravel Content	%			0.6
	T-OC	Topsoil Organic Carbon	%			0.1
	T-PH-H_2_O	Topsoil pH (H_2_O)	-			
	T-REF-BULK	Topsoil Reference Bulk Density	kg/m^3^			0.1
	T-SAND	Topsoil Sand Fraction	%			
	T-SILT	Topsoil Silt Fraction	%			0.1
	T-TEB	Topsoil TEB	-			
	T-TEXTURE	Topsoil TEXTURE	-			
	T-USDA-TEX	Topsoil USDA Texture Classification	-			**0.2**

**Table 2 biology-14-00304-t002:** The mean value (± SD) of the area under the curve (AUC) and true skill statistic (TSS) for various model algorithms.

Model Name	Model Code	AUC	TSS
Artificial neural networks model	ANN	0.8632 ± 0.1719	0.7324 ± 0.1882
Classification tree analysis model	CTA	0.8930 ± 0.0722	0.7864 ± 0.1436
Flexible discriminant analysis model	FDA	0.9274 ± 0.0429	0.7376 ± 0.0938
Generalized additive model	GAM	0.7692 ± 0.1709	0.6320 ± 0.2128
Generalized boosting model	GBM	0.9376 ± 0.0390	0.7089 ± 0.0548
Generalized linear model	GLM	0.8468 ± 0.0817	0.6936 ± 0.1635
Maximum entropy model	MaxEnt	0.9690 ± 0.0209	0.8918 ± 0.0398
Multivariate adaptive regression splines model	MARS	0.8342 ± 0.1135	0.6704 ± 0.2276
Random forest model	RF	0.9499 ± 0.0297	0.7130 ± 0.0867
Surface range envelope model	SRE	0.5352 ± 0.0498	0.1920 ± 0.0056
Ensemble model		0.9960 ±0.0641	0.9500 ±0.0610

**Table 3 biology-14-00304-t003:** Changes in the suitable habitats for *S. xylocarpa* in percent (compared to the current) under different climate scenarios. Up arrow (↑) denotes the increased case; down arrow (↓) denotes the decreased case.

Scenarios		Low Suitable Area		Moderately Suitable Area		Highly Suitable Area		Suitable Area (Moderately and Highly)	
		Area (×10^4^ km^2^)	Trend (%)	Area (×10^4^ km^2^)	Trend (%)	Area (×10^4^ km^2^)	Trend (%)	Area (×10^4^ km^2^)	Trend (%)
Last Interglacial		97.56	↑97.81	20.38	↓42.70	43.46	↑27.26	63.84	↓8.43
Middle Holocene		59.24	↑20.11	26.30	↓26.06	38.20	↑11.86	64.50	↓7.49
Current		49.32	-	35.57	-	34.15	-	69.72	-
2050s	RCP2.6	57.36	↑16.30	30.98	↓12.90	28.83	↓15.58	59.81	↓14.21
	RCP4.5	38.31	↓22.32	24.66	↓30.67	37.25	↑9.08	61.91	↓11.20
	RCP8.5	22.59	↓54.20	16.24	↓54.34	42.10	↑23.28	58.34	↓16.32
2070s	RCP2.6	47.90	↓2.88	21.48	↓39.61	38.86	↑13.79	60.34	↓13.45
	RCP4.5	75.56	↑53.20	30.06	↓15.49	29.16	↓14.61	59.22	↓15.06
	RCP8.5	64.34	↑30.45	31.44	↓11.61	41.33	↑21.02	72.77	↑4.37
The mean value of six future climate scenarios		51.01	↑3.43	25.81	↓27.44	36.26	↑6.18	62.07	↓10.97

## Data Availability

Data are contained within the article and Appendix A.

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
