# Peer review of "Evaluating Habitat Suitability for the Endangered Sinojackia xylocarpa (Styracaceae) in China Under Climate Change Based on Ensemble Modeling and Gap Analysis"

_biology, 2025, doi:10.3390/biology14030304_

Round 1

Reviewer 1 Report

Comments and Suggestions for Authors

The authors selected an endemic species in southeastern China, Sinojackia xylocarpa, as a representative, and try to forecast its potential distribution, identify the key impact factors, and analyze the conservation gaps in China’s nature reserves. They found that precipitation of the driest quarter, mean temperature and precipitation of the warmest quarter, and elevation were the four leading factors influencing its habitat suitability. Under scenarios of climate change, it is expected to shrink and shift northeastward in the future. Sure, this paper falls within the scope of this journal and the logic is fluent. The following concerns might be useful when revising the manuscript.

  • The part of the introduction is too long in its current form, which can be shortened in my view. E.g., the detail content of the introduction about this species can be moved to Methods.
  • It seemed that the amount of 21 distribution points was slightly insufficient. I think more points can be surged. Besides, published literature has been mentioned in methods, but I found no refs. If more points are available, the frequency of elevation, slope, aspect etc. can be analyzed to provide more robust results.
  • For fig.3, how to understand that precipitation of the driest quarter works for such a deciduous species? For fig.3d, illustrating elevations above 2000 m is nonsense since this species just occurs under 1000 m in my mind. Figure 4 and 5 are almost the same, the latter seems better. Fig. 8, how to understand different directions under different scenarios?
Comments on the Quality of English Language

NA

Author Response

To Reviewer 1

General comments:

"The authors selected an endemic species in southeastern China, Sinojackia xylocarpa, as a representative, and try to forecast its potential distribution, identify the key impact factors, and analyze the conservation gaps in Chinas nature reserves. They found that precipitation of the driest quarter, mean temperature and precipitation of the warmest quarter, and elevation were the four leading factors influencing its habitat suitability. Under scenarios of climate change, it is expected to shrink and shift northeastward in the future. Sure, this paper falls within the scope of this journal and the logic is fluent. The following concerns might be useful when revising the manuscript."

√ We first express gratitude to the first reviewer for his/her critical review. Then, we reply to his/her specific comments point by point.

Specific comments:

  1. "The part of the introduction is too long in its current form, which can be shortened in my view. E.g., the detail content of the introduction about this species can be moved to Methods. "

√ Thank you very much for your comment. We have partially accepted your suggestion. We have made some revisions in the section of Introduction to keep it appropriately streamlined. For example, we deleted the sentences on Line 121-123 and a reference herein.

Please see Line 63-64, 72, 101,105, 124-126,133-134, 170-175.

  1. " It seemed that the amount of 21 distribution points was slightly insufficient. I think more points can be surged. Besides, published literature has been mentioned in methods, but I found no refs. If more points are available, the frequency of elevation, slope, aspect etc. can be analyzed to provide more robust results. "

√ Thank you very much for your comment. We have accepted your suggestion, and added two citations of references in the text. In addition, a total of 104 distribution points were collected in our raw data, and after removing erroneous and duplicate points, ultimately 21 distribution records were retained. For more details, please see Table 1 which is attached below.

Please see Line 200-202.

References:

Luo, L.Q.; Luo, C. Taxonomic circumscription of Sinojackia xylocarpa (Styracaceae). J. Syst. Evol. 2011, 49(2), 163-164.

Wang, S.T.; Wu, H.; Liu, M.T.; Zhang, J.X.; Liu, J.M.; Meng, H.J.; Xu, Y.N.; Qiao, X.J.; Wei, X.Z.; Lu, Z.J.; Jiang, M.X. Community structure and dynamics of a remnant forest dominated by a plant species with extremely small population (Sinojackia huangmeiensis) in central China. Biodivers. Sci. 2018, 26(7): 749-759.

Table 1. Latitude and longitude coordinates of 104 raw data for the endangered Sinojackia xylocarpa in China.

No.

Species

Longitude(°)

Latitude(°)

1

Sinojackia xylocarpa

118.8598

32.0664

2

Sinojackia xylocarpa

118.8392

32.0594

3

Sinojackia xylocarpa

118.8412

32.0526

4

Sinojackia xylocarpa

120.2334

31.5298

5

Sinojackia xylocarpa

120.1282

30.2606

6

Sinojackia xylocarpa

112.7253

27.2706

7

Sinojackia xylocarpa

113.0385

28.1096

8

Sinojackia xylocarpa

114.3761

30.5411

9

Sinojackia xylocarpa

118.1947

30.0951

10

Sinojackia xylocarpa

108.3835

22.8600

11

Sinojackia xylocarpa

118.1164

24.4540

12

Sinojackia xylocarpa

114.1805

22.5874

13

Sinojackia xylocarpa

119.1862

31.6146

14

Sinojackia xylocarpa

114.4298

30.5503

15

Sinojackia xylocarpa

121.4797

31.2383

16

Sinojackia xylocarpa

119.0785

32.1404

17

Sinojackia xylocarpa

118.6078

32.1066

18

Sinojackia xylocarpa

118.8124

32.1452

19

20

21

Sinojackia xylocarpa

Sinojackia xylocarpa

Sinojackia xylocarpa

120.6867

31.6875

110.9125

30.0117

109.8478

27.6722

22

Sinojackia xylocarpa

110.0760

28.6694

23

Sinojackia xylocarpa

109.4886

28.5779

24

Sinojackia xylocarpa

114.5833

29.3752

25

Sinojackia xylocarpa

116.6222

29.7420

26

Sinojackia xylocarpa

115.8015

28.6577

27

Sinojackia xylocarpa

116.0011

29.5615

28

Sinojackia xylocarpa

107.2063

28.2462

29

Sinojackia xylocarpa

114.3672

30.5312

30

Sinojackia xylocarpa

111.4706

30.6565

31

Sinojackia xylocarpa

121.4510

31.1539

32

Sinojackia xylocarpa

116.0018

29.8780

33

Sinojackia xylocarpa

120.0924

30.3083

34

Sinojackia xylocarpa

120.1318

30.2679

35

Sinojackia xylocarpa

121.3840

31.1179

36

Sinojackia xylocarpa

117.2078

31.8797

37

Sinojackia xylocarpa

116.2224

39.9956

38

Sinojackia xylocarpa

116.4428

39.8806

39

Sinojackia xylocarpa

116.3529

40.0006

40

Sinojackia xylocarpa

113.3738

23.1879

41

Sinojackia xylocarpa

113.8779

22.5672

42

Sinojackia xylocarpa

114.1820

22.5822

43

Sinojackia xylocarpa

110.3125

25.0828

44

Sinojackia xylocarpa

106.9483

26.4663

45

Sinojackia xylocarpa

114.3541

36.1267

46

Sinojackia xylocarpa

114.9604

35.7864

47

Sinojackia xylocarpa

113.9402

34.7599

48

Sinojackia xylocarpa

113.5753

34.6711

49

Sinojackia xylocarpa

113.5430

34.7421

50

Sinojackia xylocarpa

114.4197

30.5544

51

Sinojackia xylocarpa

113.4676

29.8039

52

Sinojackia xylocarpa

114.3637

30.4818

53

Sinojackia xylocarpa

113.0385

28.1096

54

Sinojackia xylocarpa

113.0016

28.1373

55

Sinojackia xylocarpa

118.9649

32.1254

56

Sinojackia xylocarpa

118.8498

32.0389

57

Sinojackia xylocarpa

118.7171

32.0289

58

Sinojackia xylocarpa

118.9205

31.9041

59

Sinojackia xylocarpa

120.8761

32.0175

60

Sinojackia xylocarpa

119.1611

36.6883

61

Sinojackia xylocarpa

108.9515

34.9197

62

Sinojackia xylocarpa

109.0363

34.2152

63

Sinojackia xylocarpa

121.5590

31.2220

64

Sinojackia xylocarpa

121.4871

31.2013

65

Sinojackia xylocarpa

121.1890

31.0817

66

Sinojackia xylocarpa

121.4426

31.0341

67

Sinojackia xylocarpa

104.1365

30.7704

68

Sinojackia xylocarpa

102.7493

25.1461

69

Sinojackia xylocarpa

102.7871

25.0860

70

Sinojackia xylocarpa

120.1246

30.2598

71

Sinojackia xylocarpa

120.1986

30.2768

72

Sinojackia xylocarpa

119.7347

30.2628

73

Sinojackia xylocarpa

121.6162

29.9513

74

Sinojackia xylocarpa

121.3697

28.6343

75

Sinojackia xylocarpa

117.1851

31.9002

76

Sinojackia xylocarpa

116.0992

40.0684

77

Sinojackia xylocarpa

103.3860

29.5978

78

Sinojackia xylocarpa

114.2261

22.5857

79

Sinojackia xylocarpa

107.9616

25.2464

80

Sinojackia xylocarpa

107.8840

25.4212

81

Sinojackia xylocarpa

110.4560

19.2558

82

Sinojackia xylocarpa

114.6256

31.2958

83

Sinojackia xylocarpa

114.6468

31.2533

84

Sinojackia xylocarpa

118.8164

32.0760

85

Sinojackia xylocarpa

118.8457

32.0568

86

Sinojackia xylocarpa

118.8457

32.0568

87

Sinojackia xylocarpa

118.8060

32.0769

88

Sinojackia xylocarpa

118.8594

32.0693

89

Sinojackia xylocarpa

118.8599

32.0761

90

Sinojackia xylocarpa

120.9947

31.8452

91

Sinojackia xylocarpa

120.5908

31.2529

92

Sinojackia xylocarpa

119.7046

31.2217

93

Sinojackia xylocarpa

115.9024

29.5851

94

Sinojackia xylocarpa

121.5613

30.9530

95

Sinojackia xylocarpa

119.6445

30.8046

96

Sinojackia xylocarpa

119.6722

30.6964

97

Sinojackia xylocarpa

112.3536

28.8487

98

Sinojackia xylocarpa

112.7254

29.1870

99

Sinojackia xylocarpa

113.9125

30.5352

100

Sinojackia xylocarpa

121.2726

30.1753

101

Sinojackia xylocarpa

118.8020

32.1292

102

Sinojackia xylocarpa

117.5877

31.1281

103

Sinojackia xylocarpa

118.4850

29.9185

104

Sinojackia xylocarpa

116.0199

29.9986

  1. " For fig.3, how to understand that precipitation of the driest quarter works for such a deciduous species? For fig.3d, illustrating elevations above 2000 m is nonsense since this species just occurs under 1000 m in my mind. Figure 4 and 5 are almost the same, the latter seems better. Fig. 8, how to understand different directions under different scenarios?"

√ Thank you very much for your comments.

There are four questions posed by the second Reviewer. We have accepted these suggestions, and made revisions one by one.

(1)   Regarding the first question, we have made revisions in the sections of Result and Discussion.

(2)   Regarding the second question, we agree with the second Reviewer. So we add the following sentence in the part of “3.2. Main Environmental Factors”.

“Notably, when the elevation exceeded 1000 m, its survival probability decreased to less than 0.3.”

(3)  Regarding the third question, we agree with the second Reviewer. We have deleted the original Figure 4, and modified the sequence number of the figures in the text.

(4)   Regarding the fourth question, we have made a revision in the section of Discussion.

Please see Line 389-390, 492-501, 537-538.

Reviewer 2 Report

Comments and Suggestions for Authors

The manuscript entitled "Evaluating Habitat Suitability for the Endangered Sinojackia xylocarpa (Styracaceae) in China under Climate Change based on Ensemble Modeling and Gap Analysis " investigates the habitat suitability for the endangered Sinojackia xylocarpa in China and how climate change might impact it. The research identifies four key environmental factors that influence the distribution of Sinojackia xylocarpa: precipitation of the driest quarter, mean temperature of the warmest quarter, precipitation of the warmest quarter, and elevation. It predicts that suitable habitat for this species will decrease and become more fragmented under future climate scenarios.

In my opinion, the manuscript provides a detailed analysis of the current distribution and potential future habitat of S. xylocarpa. By identifying key environmental factors that affect its survival and distribution, the research informs conservationists and policymakers about the specific needs for protecting this endangered species. This information can help prioritize areas for conservation efforts, ensuring that actions are directed toward regions that are essential for the species' survival.

The paper would be sufficient to merit publication in Biology, though a revision is recommended, which needs to include the following points:

general suggestion: The English language could be refined a little, grammar and style polished for better readability I suggest either consulting a native speaker or running the text through a program like InstaText or similar.

Introduction:

The introduction provides a comprehensive background on climate change impacts, endangered species distributions, and the taxonomic complexity of Sinojackia xylocarpa. It also effectively introduces species distribution models (SDMs) and justifies the use of ensemble modeling with Biomod2. However, several points could be clarified or expanded to strengthen the introduction:

The authors mention previous studies (e.g., Yang et al., 2020; Feng and Zhang, 2024) but do not clearly emphasize how their work fills a gap. I suggest that the authors more explicitly highlight the novelty of their study. For example: How does using an ensemble model (Biomod2) improve upon previous single-model approaches (e.g., MaxEnt)? How does the inclusion of newly reported distribution records and updated taxonomy enhance their predictions compared to prior studies?

The introduction provides detailed taxonomic information but lacks ecological context (e.g., habitat preferences, known environmental tolerances). I suggest that the authors add a brief section on the ecological characteristics of S. xylocarpa (e.g., preferred soil types, precipitation range, temperature tolerance) to help readers understand why certain environmental variables might be important predictors.

The authors mention previous studies (e.g., Yang et al., 2020; Feng and Zhang, 2024) but do not clearly emphasize how their work fills a gap. I suggest that the authors more explicitly highlight the novelty of their study. For example: How does using an ensemble model (Biomod2) improve upon previous single-model approaches (e.g., MaxEnt)? How does the inclusion of newly reported distribution records and updated taxonomy enhance their predictions compared to prior studies?

Line 167 The authors mention assessing the conservation status of S. xylocarpa by overlaying suitable habitats with nature reserves but do not indicate how this analysis will be used for conservation recommendations. I suggest that the authors clarify how their results will contribute to conservation management strategies, such as identifying priority areas for protection or potential habitat corridors for species migration.

Materials and Methods

Line 174-186 The authors should provide more details about their field surveys, including the survey period (years and months), survey methods (e.g., transects, plot sampling, or GPS recording), and sampling effort (number of sites surveyed and geographic coverage). These details will enhance the transparency and reproducibility of their fieldwork.

Line 174-186 Explain why 1 km × 1 km grid was chosen, as different species distributions may require different scales. State how many points were removed during rarefaction.

Line 192  TableS1 lists 21 distribution points but in the Figure 21 there is 19

Line 195, Figure 2.caption  The graph shows 19 distribution points, what about the remaining two?

Line 200-202 The authors mention modeling the distribution under past, current, and future climate scenarios but do not justify why they chose specific periods (e.g., Last Interglacial, Middle Holocene, 2050s, and 2070s). I recommend that the authors briefly explain why these periods were selected and how they contribute to understanding distribution dynamics and conservation implications

Line 239 Table1 The authors included multiple soil properties, but the contributions are very low (mostly ≤0.7%). Please clarify why so many low-contribution soil variables were retained? Was their inclusion based on ecological relevance or prior knowledge?

Line 245-258 The modeling workflow using the Biomod2 package for species distribution modeling appears methodologically sound and follows standard practices for ensemble modeling, with appropriate model selection criteria (AUC and TSS thresholds) and standard ensemble modeling procedures. However, I have few suggestions and and clarifications that may improve the clarity and robustness of the methods section:

  • justification for model selection thresholds

The authors selected models with AUC > 0.8 and TSS > 0.7 to build the ensemble. However, their own classification (Wang et al., 2024; Liu et al., 2024) defines "excellent" models as TSS > 0.8. Why did the authors use a more lenient threshold for TSS compared to AUC? I suggest clarifying the rationale for using different thresholds.

  • number of Pseudo-Absence Points

The authors generated 1000 pseudo-absence points during the modeling process. Since pseudo-absence selection can significantly influence model performance, I recommend explaining the rationale for choosing 1000 points.

Line 268-270 Explain why the moderately and highly suitable areas were grouped as suitable habitat. Was this decision based on biological knowledge of S. xylocarpa or purely numerical classification? A brief rationale would add clarity.

Results

Line 277-279 Although the threshold values for AUC and TSS are explained in detailed in the Materials and Methods section, it is not necessary to repeat them in detail here. However, you could briefly reference the criteria to maintain clarity without redundancy. For example: "Following the selection criteria described in the Materials and Methods section, we included models with AUC > 0.8 and TSS > 0.7 in the ensemble model."

Line 286 Provide a brief explanation of why the ensemble model performed better than individual models. For instance: "The ensemble model's superior performance likely results from combining predictions from multiple algorithms, which reduces individual model biases and enhances predictive accuracy."

Line 289 Table 2. Include standard deviations, confidence intervals, or other measures of variability for AUC and TSS scores to give readers a sense of model reliability. Ensure that Table 2 is well-organized with clear labels (e.g., Model Type, AUC, TSS) and highlight the ensemble model results for easy comparison. Align left the text within the “Model name” and “Model code” columns.

Line 291-321

The shift from current to past periods (Last Interglacial, Middle Holocene) happens abruptly. Add a brief transition sentence to guide the reader. For example: "The contribution of environmental factors varied across different periods, highlighting changes in habitat suitability over time."

To improve readability, the structure of the 3.2. section should be divided it into the paragraphs related to current period contributions, contributions in the past periods (Last Interglacial and Middle Holocene) and Response curves and habitat suitability

Line 303 Emphasize shifts in key environmental drivers over time. For instance: "Notably, Bio17 (precipitation of the driest quarter), which is the dominant factor in the current period, ranked second during both the Last Interglacial and Middle Holocene periods, suggesting a shift in climatic drivers of habitat suitability over time."

Line 307 Briefly explain the ecological meaning of the response curves (e.g., thresholds for suitability). For example: "The response curves indicate ecological thresholds, where probability of species presence changes non-linearly in response to key environmental factors."

Additionally, state whether the relationships are linear, unimodal, or asymptotic, as it helps in interpreting the species' ecological preferences.

Line 310 Reference figures more effectively: "As shown in Figure 3a, the probability of S. xylocarpa presence increases sharply with precipitation of the driest quarter, stabilizes, and then decreases."

Line 317 Report ranges clearly: Instead of: "S. xylocarpa grew well when the precipitation of the warmest quarter was between 417 mm and 763 mm" Use: "The suitable precipitation range for the warmest quarter was 417–763 mm"

Line 322 The graphs in the Figure 3 needs to be sharper, the axes titles are barely readable. Maybe it would help to increase the pictures resolution. Also, increase the numbers on the axes. the letters are too small and therefore difficult to read.

Line 332 To enhance readability and ensure clarity for a wider audience, I recommend expressing areas in standard numerical format (e.g., 697,200 km²) instead of scientific notation (e.g., 69.72 × 10⁴ km²). This format is more intuitive and allows for easier comparison of values, especially for readers less familiar with scientific notation. Also see lines 334, 338, 340, 355, 361, 374, 382, 384.

In the Table 3. keep such formatting due to the physical limitations of the table columns

Line 363 I suggest this sentence: "The results show a continuous contraction and fragmentation of suitable habitats from the Last Interglacial to the Middle Holocene, indicating increasing environmental constraints for S. xylocarpa over time."

Since fragmentation is mentioned, briefly explain its significance: "In the Middle Holocene, increased fragmentation suggests reduced connectivity between habitats, which could have impacted species dispersal and survival."

Line 407-410  Change the sentence "The centroid of S. xylocarpa’s suitable habitat shifted southwest from the Last Interglacial to the Middle Holocene and then southeast to the present. In future projections, the centroid generally moves northeast under all RCP scenarios, indicating a shift toward higher latitudes as the climate warms."

Provide ecological context explaining what the centroid shift implies for species survival. For example: "This northeastward shift reflects a common response of species to climate change, moving toward cooler regions as temperatures rise. However, such shifts may result in habitat loss if suitable areas become fragmented or unavailable."

Discussion

My general suggestion to refine the English language throughout the manuscript to improve readability, particularly relates to the discussion section, where clarity and coherence are especially important.

For example, correct “The outcomes from our ensemble confirmed this” (Line 417) to “Our ensemble model results confirmed this observation.”,  change “...the results displayed that the prediction is generally consistent” (Line 424). to “...showed that the prediction was generally consistent”, change “Our results indicate that S. xylocarpa have much larger suitable habitats” (Line 474). to “Our results indicate that S. xylocarpa has a much larger suitable habitat.” , etc.

To ensure consistency and improve readability, I recommend converting all areas currently expressed in scientific notation (e.g., 52.25 × 10⁴ km²) to standard numerical format (e.g., 522,500 km²) throughout the text

Section 4.2.

This section effectively identifies the major environmental drivers and compares findings with previous research, but it needs to be improved a little, here are some suggestions:

grammar and style

Line 433  Change“...Bio17 has the largest contribution rate...” to “Bio17 contributed 61.0%, Bio10 9.6%, and Bio18 8.9%”, or the part “which is much larger than of Bio10 (9.6%)...” to “which is much larger than that of Bio10 (9.6%)...” etc

(Line 436)Provide a clearer comparison with Zhu et al. (2024). For example: "Unlike Zhu et al. (2024), who found temperature-related variables to be dominant, our results emphasize the importance of precipitation, likely due to the larger sample size (21 distribution points vs. 2)."

Section 4.3.

Expand the discussion on why the model shows suitable areas in Taiwan despite the absence of records. Clarify the discrepancy with Jin et al. (2023).

Section 4.5

Propose clear conservation strategies and recommend specific actions such as:

In Situ conservation: Expand existing nature reserves in highly suitable areas.

Ex Situ conservation: Develop seed banks and botanical gardens to maintain genetic diversity.

Restoration: Reconnect fragmented habitats through ecological corridors.

Comments on the Quality of English Language

The English language could be refined a little, grammar and style polished for better readability I suggest either consulting a native speaker or running the text through a program like InstaText or similar

Author Response

To Reviewer 2

General comments:

"The manuscript entitled "Evaluating Habitat Suitability for the Endangered Sinojackia xylocarpa (Styracaceae) in China under Climate Change based on Ensemble Modeling and Gap Analysis " investigates the habitat suitability for the endangered Sinojackia xylocarpa in China and how climate change might impact it. The research identifies four key environmental factors that influence the distribution of Sinojackia xylocarpa: precipitation of the driest quarter, mean temperature of the warmest quarter, precipitation of the warmest quarter, and elevation. It predicts that suitable habitat for this species will decrease and become more fragmented under future climate scenarios.

In my opinion, the manuscript provides a detailed analysis of the current distribution and potential future habitat of S. xylocarpa. By identifying key environmental factors that affect its survival and distribution, the research informs conservationists and policymakers about the specific needs for protecting this endangered species. This information can help prioritize areas for conservation efforts, ensuring that actions are directed toward regions that are essential for the species' survival.

The paper would be sufficient to merit publication in Biology, though a revision is recommended, which needs to include the following points:

general suggestion: The English language could be refined a little, grammar and style polished for better readability. I suggest either consulting a native speaker or running the text through a program like InstaText or similar."

√ We first express gratitude to the second reviewer for his/her critical review. Then, we reply to his/her specific comments one by one. In addition, we invite Dr. Yu Shengxiang, who is a professor from Institute of Botany, Chinese Academy of Sciences, to help us to polish the English language.

Specific comments:

  1. "Introduction:

The introduction provides a comprehensive background on climate change impacts, endangered species distributions, and the taxonomic complexity of Sinojackia xylocarpa. It also effectively introduces species distribution models (SDMs) and justifies the use of ensemble modeling with Biomod2. However, several points could be clarified or expanded to strengthen the introduction:

The authors mention previous studies (e.g., Yang et al., 2020; Feng and Zhang, 2024) but do not clearly emphasize how their work fills a gap. I suggest that the authors more explicitly highlight the novelty of their study. For example: How does using an ensemble model (Biomod2) improve upon previous single-model approaches (e.g., MaxEnt)? How does the inclusion of newly reported distribution records and updated taxonomy enhance their predictions compared to prior studies?"

√ Thank you very much for your comments.

The two previous studies (e.g., Yang et al., 2020; Feng and Zhang, 2024) were carried out at the level of genus (i.e. Sinojackia), but our study addressed the potential distribution of endangered S. xylocarpa at the level of species. Our study is different from the previous ones in that it adopts the ensemble model, and combines environmental variables with the inclusion of the updated taxonomy and newly reported distribution records of this species.

Even so, we have made a revision according to the suggestion of the second reviewer.

Please see Line 153-161.

2."The introduction provides detailed taxonomic information but lacks ecological context (e.g., habitat preferences, known environmental tolerances). I suggest that the authors add a brief section on the ecological characteristics of S. xylocarpa (e.g., preferred soil types, precipitation range, temperature tolerance) to help readers understand why certain environmental variables might be important predictors."

√ Thank you very much for your comment.

This question is similar to Question10 below. To the best of our knowledge, there are few reports on the ecological characteristics of this tree species. The probable reasons are as follows:

Firstly, S. xylocarpa is endemic to China. This species was established by Prof. Hu Xiansu in 1929. For a long time, it is thought that S. xylocarpa is only distributed in mountainous area of southern Jiangsu, east China. Namely, it is considered as a tree species endemic to Jiangsu Province. More recently, there have successively been some new distribution localities reported in China over the several decades. Due to its late discovery and restricted distribution, little is known about the ecological characteristics of S. xylocarpa.

According to the reviewer's comments, we have supplemented a reference and made a revision in the section of Introduction. We add the following sentence on Line 83-84.

S. xylocarpa is a light-demanding tree, and grows well in a warm humid climate.  Moreover, it pays little emphasis on soil (Sheng et al., 2012).”

Please see Line 83-84 for more details.

References:

Sheng, N.; Li, B.J.; Xiong, Y.N.; Lu, C.G.; Lin, L. Garden Ornamental Trees in East China. Shanghai Science and Technology Press: Shanghai, China, 2012, 156-157.

  1. "The authors mention previous studies (e.g., Yang et al., 2020; Feng and Zhang, 2024) but do not clearly emphasize how their work fills a gap. I suggest that the authors more explicitly highlight the novelty of their study. For example: How does using an ensemble model (Biomod2) improve upon previous single-model approaches (e.g., MaxEnt)? How does the inclusion of newly reported distribution records and updated taxonomy enhance their predictions compared to prior studies?"

√ Thank you very much for your comment. We have made a revision according to the suggestion of the second reviewer.

In addition, this question (i.e. Question 3) is the same as Question1. Please see the answer to question 1.

  1. "Line 167 The authors mention assessing the conservation status of S. xylocarpa by overlaying suitable habitats with nature reserves but do not indicate how this analysis will be used for conservation recommendations. I suggest that the authors clarify how their results will contribute to conservation management strategies, such as identifying priority areas for protection or potential habitat corridors for species migration."

√ Thanks for your critical review. We have accepted the suggestion, and made revisions in the sections of Introduction and Discussion, respectively.

Please see Line 174-175, 620- 623.

  1. "Materials and Methods

Line 174-186 The authors should provide more details about their field surveys, including the survey period (years and months), survey methods (e.g., transects, plot sampling, or GPS recording), and sampling effort (number of sites surveyed and geographic coverage). These details will enhance the transparency and reproducibility of their fieldwork."

  Thank you very much for your comment. We have accepted your suggestion, and made a revision.

We provide more details about field investigation information in the section of Materials and Methods. The details are as follows:

“Investigating in the field: From the spring of 2022 to the autumn of 2024, we conducted a survey of the wild population of S. xylocarpa in Anhui (i.e. Xuancheng, Wuhu, Hefei), Jiangsu (i.e. Nanjing, Changzhou, Zhenjiang), Zhejiang (i.e. Huzhou, Hangzhou, Ningbo), and other provinces in eastern China to determine its distribution. Simultaneously, we located the S. xylocarpa populations with GPS and recorded their geographical coordinates (i.e. latitude and longitude).”

Please see Line 183 -191 for more details.

  1. "Line 174-186 Explain why 1 km × 1 km grid was chosen, as different species distributions may require different scales. State how many points were removed during rarefaction."

√ Thank you very much for your comments.

As an endemic and endangered tree species in China, S. xylocarpa has few distribution points, and therefore more distribution points can be retained by selecting high resolution (i.e. 1 km × 1 km grid). More importantly, finer scale data are considered to better reflect climatic conditions experienced by species (Franklin et al., 2013). The same practice is seen in Wang et al. (2024) and Liu et al. (2024). Besides, after eliminating erroneous and duplicate distribution records, we collected 22 natural distribution points of S. xylocarpa, and just one point was removed during rarefaction.

Please see Line 199 - 201.

References:

Franklin, J.; Davis, F.W.; Ikegami, M.; Syphard, A.D.; Flint, L.; Flint, A.L.; Hannah, L. Modeling plant species distributions under future climates: how fine scale do climate projections need to be? Glob. Chang. Biol. 2013, 19(2): 473-483.

Liu, T.; Cai, H.W.; Zhang, G.F. Assessment of climate change impacts on the distribution of endangered and endemic Changnienia amoena (Orchidaceae) using ensemble modeling and gap analysis in China. Ecol. Evol. 2024, 14(11), e70636.

Wang, H.R.; Zhi, F.Y.; Zhang, G.F. Predicting impacts of climate change on suitable distribution of critically endangered tree species Yulania zenii (W. C. Cheng) D. L. Fu in China. Forests 2024, 15(5), 883.

  1. "Line 192  TableS1 lists 21 distribution points but in the Figure 2 there is 19"

√ Thank you very much for your comment.

Because some of the 21 points are close to each other, there are overlaps in Figure 2. Actually, all of the cleaned data are shown in Figure 2. There are 21 distribution points for S. xylocarpa in total.

  1. "Line 195, Figure 2.caption  The graph shows 19 distribution points, what about the remaining two?"

√ Thanks very much for your comment.

One of the remaining two points is partially overlapped with the other one because they are close to each other on the map. In fact, Figure 2 shows 21 distribution points. For more details, please see the attached Table S1.

  1. "Line 200-202 The authors mention modeling the distribution under past, current, and future climate scenarios but do not justify why they chose specific periods (e.g., Last Interglacial, Middle Holocene, 2050s, and 2070s). I recommend that the authors briefly explain why these periods were selected and how they contribute to understanding distribution dynamics and conservation implications"

Thank you very much for your comments.

We have accepted the suggestion, and explained why these periods were selected and how they contributed to understanding distribution dynamics and conservation implications.

Firstly, Last Interglacial (LIG) was one of the warmest interglacial periods in the past 800,000 years, with its warming closely related to changes in incoming solar radiation at the top of the atmosphere caused by variations in Earth's orbital parameters (Turney et al., 2020). During the LIG, vegetation zones in the Northern Hemisphere expanded northward as a whole (Pausata et al., 2020). Given that the climatic characteristics of the LIG are similar to the long-term trends of future climate change projections (Gulev et al., 2021), the LIG has become one of the key warm periods for paleoclimate research both domestically and internationally.

Secondly, the Middle Holocene (MH) corresponds roughly to a period between 5,000 and 7,000 years BP: This period is the most recent warm period on an orbital scale, with changes in Earth's orbital parameters leading to increased summer insolation in the extratropical regions of the Northern Hemisphere. It has a similar warming effect on climate as the modern increase in atmospheric CO₂ (Berger et al., 1978; Shin et al., 2006). Understanding processes and mechanisms of climate change during the MH can help elucidate the impact and contribution of natural warming in the climate change process, providing an important scientific basis for predicting future climate change trends and formulating climate change response strategies (Chen et al., 2023).

Therefore, we choose LIG and MH as the two periods in the past.

Generally, there are four future periods including 2030s, 2050s, 2070S and 2090s. We selected 2050s and 2070s as future scenarios because these periods cover the near and medium-term impacts of climate change. By predicting the distribution during these periods, we can assess the potential threats of future climate change to S. xylocarpa, thus providing a basis for conservation measures.

Therefore, we choose 2050s and 2070s as the two periods in the future.

In addition, we add two citations herein.

Please see Line 220-221.

References:

Berger, A.L. Long-term variations of caloric insolation resulting from the Earth's orbital elements. Quat. Res. 1978, 9(2): 139-167.

Chen, F.H.; Duan, Y.W.; Hao, S.; Chen, J.; Feng, X.P.; Hou, J.Z.; Cao, X.Y.; Zhang, X.; Zhou, T.J. Holocene thermal maximum mode versus the continuous warming mode: Problems of data-model comparisons and future research prospects. Sci. China Earth Sci. 2023, 66(8): 1683–1701.

Gulev, S.K.; Thorne, P.W.; Ahn, J.; et al. Changing state of the climate system[R]//Masson-Delmotte, V.; Zhai, P.; Pirani, S.; et al. Climate change 2021: the physical science basis. Contribution of working group I to the Sixth Assessment Report of the Intergovernmental Panel on Climate Change. Cambridge: Cambridge University Press, 2021: 287-422.

Jiang, N.X.; Yan, Q.; Wang, H.J. Transient and time-slice simulations of global climate change during the Last Interglacial: Model-model and model-data comparisons. Earth Sci. Front. 2024, 31(1): 486-499.

Pausata, F.S.R.; Gaetani, M.; Messori, G.; Berg, A.; Souza, D.M.D.; Sage, R.F.; deMenocal, P.B. The greening of the Sahara: past changes and future implications. One Earth 2020, 2(3): 235-250.

Shin, S.I.; Sardeshmukh, P.D.; Webb, R.S.; Oglesby, R.J. Understanding the mid-Holocene climate. J. Clim. 2006, 19(12): 2801-2817.

Turney, C.S.M.; Jones, R.T.; Mckay, N.P.; Sebille, E.V.; Thomas, Z.A.; Hillenbrand, C.; Fogwill, C.J. A global mean sea-surface temperature dataset for the Last Interglacial (129-116 kyr) and contribution of thermal expansion to sea-level change. Earth Syst. Sci. Data. 2020, 12(4): 3341-3356.

  1. "Line 239 Table1 The authors included multiple soil properties, but the contributions are very low (mostly ≤0.7%). Please clarify why so many low-contribution soil variables were retained? Was their inclusion based on ecological relevance or prior knowledge?"

√ Thank you very much for your comment.

Firstly, S. xylocarpa is endemic to China. This species was found in 1929. Up to now, there have successively been some new distribution localities reported in China over the several decades. Due to its late discovery and restricted distribution, little is known about the ecological characteristics of S. xylocarpa, especially soil properties. Hence, we selected the 17 soil variables while modeling. In addition, such a practice is also seen in other researches. Feng et al. (2022) introduced all soil variables when modeling the distribution of Ostrya rehderiana, an endangered tree endemic to China. Therefore, we do not make a revision for the time being. Anyway, we appreciate your comments, and we will consider this issue in the future study.

References:

Feng, L.; Sun, J.J.; El-Kassaby, Y.A.; Yang, X.Y.; Tian, X.N.; Wang, T.L. Predicting potential habitat of a plant species with small populations under climate change: Ostrya rehderiana. Forests 2022, 13(1): 129.

11."Line 245-258 The modeling workflow using the Biomod2 package for species distribution modeling appears methodologically sound and follows standard practices for ensemble modeling, with appropriate model selection criteria (AUC and TSS thresholds) and standard ensemble modeling procedures. However, I have few suggestions and and clarifications that may improve the clarity and robustness of the methods section:

justification for model selection thresholds

The authors selected models with AUC > 0.8 and TSS > 0.7 to build the ensemble. However, their own classification (Wang et al., 2024; Liu et al., 2024) defines "excellent" models as TSS > 0.8. Why did the authors use a more lenient threshold for TSS compared to AUC? I suggest clarifying the rationale for using different thresholds."

√ Thank you very much for your comment. We have accepted your suggestion, and made a revision.

Firstly, as a rule, the AUC value typically ranged from 0 to 1, and a value closer to 1 indicated higher accuracy. It can be categorized into five groups: excellent (0.9-1.0), good (0.8-0.9), fair (0.7-0.8), poor (0.6-0.7), and failing (0.5-0.6) (Singh et al., 2021).

Secondly, TSS values varied between -1 and +1, with values closer to 1 indicating superior performance, while values closer to or below 0 suggested inferior performance. However, there is no consensus on the threshold value of TSS when evaluating model performance. Some scholars propose that models can also be divided into five groups in terms of TSS: excellent (TSS > 0.8), good (0.6-0.8), fair (0.4-0.6), poor (0.2-0.4), and failing (TSS < 0.2) (Wang et al., 2023).

Thirdly, in this study, we used the values of both AUC and TSS to evaluate the performance of the ensemble model. At present, there is no consensus on the threshold values of AUC and TSS when evaluating ensemble model performance, so herein we followed the method of ensemble model evaluation for Emmenopterys henryi Oliv., an endangered tree endemic to China.

Please see Line 277-282.

References:

Singh, M.; Arunachalam, R.; Kumar, L. Modeling potential hotspots of invasive Prosopis juliflora (Swartz) DC in India. Ecol. Inform. 2021, 64, 101386.

Wang, Z.W.; Yin, J.; Wang, X.; Chen, Y.; Mao, Z.K.; Lin, F.; Gong, Z.Q.; Wang, X.G. Habitat suitability evaluation of invasive plant species Datura stramonium in Liaoning Province: Based on Biomod2 combination model. Chin. J. Appl. Ecol. 2023, 34, 1272-1280.

  1. "number of Pseudo-Absence Points

The authors generated 1000 pseudo-absence points during the modeling process. Since pseudo-absence selection can significantly influence model performance, I recommend explaining the rationale for choosing 1000 points."

Thank you very much for your comment.

At present, there is still no consensus on how many pseudo-absence points should be used in species distribution modeling. According to the existing literature, the number of pseudo-absences can be set as 100 (i.e. Luke et al., 2019), 1000 (i.e. Chang et al., 2024; Zhang et al., 2023), or even 10000 (i.e. Mazzolari et al., 2020).

According to the literature analysis, we tend to think that for a specific species, the number of pseudo-absences primarily depends on modelling techniques, species prevalence, and the number of presence points used in SDMs (Liu et al., 2019). Furthermore, the method for selecting pseudo-absences (i.e. random vs. environmentally or spatially stratified) may take an effect on the predictive accuracy of used model (Barbet et al., 2012).

There are 21 distribution points of S. xylocarpa in this study. For species with fewer distribution points, most researchers usually adopt 1000 pseudo-absence points. Some examples are the endangered Taxus cuspidata (Chang et al., 2024), Zelkova carpinifolia (Koç et al., 2024), and three tree species from Rosaceae (Guo et al., 2024). In addition, when the pseudo-absence points are randomly generated, the model will have higher accuracy (Barbet et al., 2012).

 Furthermore, the outcomes from the final models in this study confirm this. For example, the AUC value and TSS value were all above 0.9, indicating that these models have reached an excellent level. Therefore, we randomly generated 1000 pseudo-absence points during the modeling process.

Please see Line 284-285.

References:

Barbet-Massin, M.; Jiguet, F.; Albert, H. C.; Thuiller, W. Selecting pseudo‐absences for species distribution models: how, where and how many? Methods Ecol. Evol. 2012, 3: 327-338.

Chang, B.; Huang, C.; Chen, B.; Wang, Z.W.; He, X.Y.; Chen, W.; Huang, Y.Q.; Zhang, Y.; Yu, S. Predicting the potential distribution of Taxus cuspidata in northeastern China based on the ensemble model. Ecosphere 2024, 15(8), e4965.

Guo, F.; Yang, Y.; Gao, G. Climate change impact on three important species of wild fruit forest ecosystems: assessing habitat loss and climatic niche shift. Forests 2024, 15, 1281.

Koç, D.E.; Beyza, U.; Demet, B. Effect of climate change on the habitat suitability of the relict species Zelkova carpinifolia Spach using ensembled species distribution modelling. Sci. Rep. 2024, 14, 27967.

Liu, C.R.; Graeme, N.; Matt, W. The effect of sample size on the accuracy of species distribution models: considering both presences and pseudo-absences or background sites, Ecography 2019, 42, 535–548.

Luke, K.B.; Robertson, M. P.; Barker, N. P. Range contraction to a higher elevation: the likely future of the montane vegetation in South Africa and Lesotho. Biodivers. Conserv. 2019, 28, 131-153.

Mazzolari, A.C.; Millan, E.N.; Bringa, E.M.; Vázquez, D.P. Modeling habitat suitability and spread dynamics of two invasive rose species in protected areas of Mendoza, Argentina. Ecol. Complex. 2020, 44, 100868.

Zhang, Y.; Jiang, X.; Lei, Y.; Wu, Q.L.; Liu, Y.H.; Shi, X.W. Potentially suitable distribution areas of Populus euphratica and Tamarix chinensis by MaxEnt and random forest model in the lower reaches of the Heihe River, China. Environ. Monit. Assess. 2023, 195(12), 1519.

  1. "Line 268-270 Explain why the moderately and highly suitable areas were grouped as suitable habitat. Was this decision based on biological knowledge of S. xylocarpa or purely numerical classification? A brief rationale would add clarity. "

Thank you very much for your comment. We have adopted it and made revisions.

Generally, the potential suitable area for plant species is usually divided into four categories (i.e. unsuitable, low suitability, moderately suitable, highly suitable) (Watling et al., 2015; Zhou et al., 2023; Wang et al., 2024). There are two methods to classify the projected area:

(1) Numerical classification. The potential area is divided into four equal portions in size. Namely, they are unsuitable area (0-0.25), low suitable area (0.25-0.5), moderately suitable area (0.5-0.75), and highly suitable area (0.75-1) (Lu et al., 2022).

(2) Threshold classification. It is more appropriate for endangered species to reclassify the model results on the basis of the "test sensitivity and specificity threshold" compared to non-endangered species. In our study, we followed the approach of Liu et al. (2013) to reclassify the potential area for S. xylocarpa, which is an endangered tree species endemic to China. We reclassified the model results on the basis of the "test sensitivity and specificity threshold" (0.19) when only presence data were available (Liu et al., 2013), then divided its habitat suitability into four categories in light of such a threshold value. They were unsuitable area (0.00-0.19), low suitable area (0.19-0.46), moderately suitable area (0.46-0.73), and highly suitable area (0.73-1.00).

Moreover, considering the small size of its highly suitable area, we merged the moderately and highly suitable areas into suitable area for S. xylocarpa. Such a practice has been widely used in the distribution prediction for endangered plants (Dhyani et al., 2021; Wei et al., 2024).

Please see Line 297-298.

References:

Dhyani, A.; Kadaverugu, R.; Nautiyal, B.P.; Nautiyal, M.C. Predicting the potential distribution of a critically endangered medicinal plant Lilium polyphyllum in Indian Western Himalayan Region. Reg. Environ. Change. 2021, 21(2), 30.

Liu, C.R.; White, M.; Newell, G. Selecting thresholds for the prediction of species occurrence with presence‐only data. J. Biogeogr. 2013, 40(4), 778-789.

Lu, X.; Jiang, R.Y.; Zhang, G.F. Predicting the potential distribution of four endangered holoparasites and their primary hosts in China under climate change. Front. Plant. Sci. 2022, 13, 942448.

Wang, H.R.; Zhi, F.Y.; Zhang, G.F. Predicting impacts of climate change on suitable distribution of critically endangered tree species Yulania zenii (W. C. Cheng) D. L. Fu in China. Forests 2024, 15(5), 883.

Watling, J.I.; Brandt, L.A.; Bucklin, D.N.; Fujisaki, I.; Mazzotti, F.J.; Romanach, S.S.; Speroterra, C. Performance metrics and variance partitioning reveal sources of uncertainty in species distribution models. Ecol. Modell. 2015, 309, 48-59.

Wei, L.; Wang, G.; Xie, C.; Gao, Z.; Huang, Q.; Jim, C.Y. Predicting suitable habitat for the endangered tree Ormosia microphylla in China. Sci. Rep. 2024, 14(1), 10330.

Zhou, Y.R.; Lu, X.; Zhang, G.F. Potentially differential impacts on niche overlap between Chinese endangered Zelkova schneideriana and its associated tree species under climate change. Front. Ecol. Evol. 2023, 11, 1218149.

  1. "Results

Line 277-279 Although the threshold values for AUC and TSS are explained in detailed in the Materials and Methods section, it is not necessary to repeat them in detail here. However, you could briefly reference the criteria to maintain clarity without redundancy. For example: "Following the selection criteria described in the Materials and Methods section, we included models with AUC > 0.8 and TSS > 0.7 in the ensemble model.""

Thank you very much for your comment. We have adopted it and made a revision.

Please see Line 330-332.

  1. "Line 286 Provide a brief explanation of why the ensemble model performed better than individual models. For instance: "The ensemble model's superior performance likely results from combining predictions from multiple algorithms, which reduces individual model biases and enhances predictive accuracy." "

Thank you very much for your comment. We have adopted it and made a revision.

Please see Line 342-344.

  1. "Line 289 Table 2. Include standard deviations, confidence intervals, or other measures of variability for AUC and TSS scores to give readers a sense of model reliability. Ensure that Table 2 is well-organized with clear labels (e.g., Model Type, AUC, TSS) and highlight the ensemble model results for easy comparison. Align left the text within the “Model name” and “Model code” columns."

Thank you very much for your comment. We have adopted it and made a revision.

Please see Line 347-348.

  1. "Line 291-321

The shift from current to past periods (Last Interglacial, Middle Holocene) happens abruptly. Add a brief transition sentence to guide the reader. For example: "The contribution of environmental factors varied across different periods, highlighting changes in habitat suitability over time." "

Thank you very much for your comment. We have adopted it and made a revision.

Please see Line 366-367.

  1. "To improve readability, the structure of the 3.2. section should be divided it into the paragraphs related to current period contributions, contributions in the past periods (Last Interglacial and Middle Holocene) and Response curves and habitat suitability"

√ Thank you very much for your comment. We have adopted it and made a revision.

Please see Line 350-367.

  1. "Line 303 Emphasize shifts in key environmental drivers over time. For instance: "Notably, Bio17 (precipitation of the driest quarter), which is the dominant factor in the current period, ranked second during both the Last Interglacial and Middle Holocene periods, suggesting a shift in climatic drivers of habitat suitability over time.""

√ Thank you very much for your comment. We have adopted it and made a revision.

Please see Line 361-363.

  1. "Line 307 Briefly explain the ecological meaning of the response curves (e.g., thresholds for suitability). For example: "The response curves indicate ecological thresholds, where probability of species presence changes non-linearly in response to key environmental factors.""

 "Additionally, state whether the relationships are linear, unimodal, or asymptotic, as it helps in interpreting the species' ecological preferences."

Thank you very much for your valuable comment. We have accepted your suggestion and made a revision. We add the following sentence on Line 362-364.

“As a result, the response curves of Bio17 (in Fig. 3a) and Bio18 (in Fig. 3c) present a unimodal pattern while the response curves of Bio10 (in Fig. 3b) and elevation (in Fig. 3d) present an asymptotic pattern.”

Please see Line 372-374, 390-392.

  1. "Line 310 Reference figures more effectively: "As shown in Figure 3a, the probability of S. xylocarpa presence increases sharply with precipitation of the driest quarter, stabilizes, and then decreases.""

√ Thank you very much for your comment. We have adopted it and made a revision.

Please see Line 376-377.

  1. "Line 317 Report ranges clearly: Instead of: "S. xylocarpa grew well when the precipitation of the warmest quarter was between 417 mm and 763 mm" Use: "The suitable precipitation range for the warmest quarter was 417–763 mm""

√ Thank you very much for your comment. We have adopted it and made a revision.

Please see Line 383-384.

  1. "Line 322 The graphs in the Figure 3 needs to be sharper, the axes titles are barely readable. Maybe it would help to increase the pictures resolution. Also, increase the numbers on the axes. the letters are too small and therefore difficult to read."

Thank you very much for your comment.

In this study, the resolution of Figure 3 is high (i.e. 12460 × 8010 pixels), which meets the requirements of the journal. The possible reason is that the figure becomes blurred after being inserted in the text. We have uploaded the original figure 3 on the journal website.

Please see Line 393.

  1. "Line 332 To enhance readability and ensure clarity for a wider audience, I recommend expressing areas in standard numerical format (e.g., 697,200 km²) instead of scientific notation (e.g., 69.72 × 10⁴ km²). This format is more intuitive and allows for easier comparison of values, especially for readers less familiar with scientific notation. Also see lines 334, 338, 340, 355, 361, 374, 382, 384."

 "In the Table 3. keep such formatting due to the physical limitations of the table columns"

√ Thank you very much for your comment. We have adopted it and made revisions.

Please see Line 403, 405, 409, 411, 430, 437, 458, 466, 468.

  1. "Line 363 I suggest this sentence: "The results show a continuous contraction and fragmentation of suitable habitats from the Last Interglacial to the Middle Holocene, indicating increasing environmental constraints for S. xylocarpa over time." "

"Since fragmentation is mentioned, briefly explain its significance: "In the Middle Holocene, increased fragmentation suggests reduced connectivity between habitats, which could have impacted species dispersal and survival." "

√ Thank you very much for your comments. We have accepted the suggestions and made revisions.

Please see Line 439-445.

  1. "Line 407-410  Change the sentence "The centroid of S. xylocarpa’s suitable habitat shifted southwest from the Last Interglacial to the Middle Holocene and then southeast to the present. In future projections, the centroid generally moves northeast under all RCP scenarios, indicating a shift toward higher latitudes as the climate warms." "

"Provide ecological context explaining what the centroid shift implies for species survival. For example: "This northeastward shift reflects a common response of species to climate change, moving toward cooler regions as temperatures rise. However, such shifts may result in habitat loss if suitable areas become fragmented or unavailable." "

√ Thank you very much for your comment. We have accepted the suggestion and made revisions.

Please see Line 492-501.

  1. "Discussion

My general suggestion to refine the English language throughout the manuscript to improve readability, particularly relates to the discussion section, where clarity and coherence are especially important. "

"For example, correct “The outcomes from our ensemble confirmed this” (Line 417) to “Our ensemble model results confirmed this observation.”,  change “...the results displayed that the prediction is generally consistent” (Line 424). to “...showed that the prediction was generally consistent”, change “Our results indicate that S. xylocarpa have much larger suitable habitats” (Line 474). to “Our results indicate that S. xylocarpa has a much larger suitable habitat.” , etc. "

√ Thank you very much for your comment. We have accepted the suggestion and made revisions.

Please see Line 508-509, 516, 573-575.

  1. "To ensure consistency and improve readability, I recommend converting all areas currently expressed in scientific notation (e.g., 52.25 × 10⁴ km²) to standard numerical format (e.g., 522,500 km²) throughout the text "

√ Thank you very much for your comment. We have accepted the suggestion and made revisions.

Please see Line 552, 578, 579, 604.

  1. "Section 4.2.

This section effectively identifies the major environmental drivers and compares findings with previous research, but it needs to be improved a little, here are some suggestions:

grammar and style

Line 433  Change“...Bio17 has the largest contribution rate...” to “Bio17 contributed 61.0%, Bio10 9.6%, and Bio18 8.9%”, or the part “which is much larger than of Bio10 (9.6%)...” to “which is much larger than that of Bio10 (9.6%)...” etc"

√ Thank you very much for your comment. We have accepted the suggestion and made revisions.

Please see Line 526-527.

  1. "(Line 436)Provide a clearer comparison with Zhu et al. (2024). For example: "Unlike Zhu et al. (2024), who found temperature-related variables to be dominant, our results emphasize the importance of precipitation, likely due to the larger sample size (21 distribution points vs. 2)." "

√ Thank you very much for your comment. We have accepted the suggestion and made revisions.

Please see Line 529-531.

  1. "Section 4.3.

Expand the discussion on why the model shows suitable areas in Taiwan despite the absence of records. Clarify the discrepancy with Jin et al. (2023). "

√ Thank you very much for your comments. We have accepted the two suggestions.

As for the first suggestion, here is our understanding.

For a plant species, its suitable habitat projected by species distribution model only indicates that this species is likely to occur in such an area. In fact, its actual geographical range depends largely on its suitable habitat, as well as on its origin and evolution, propagule dispersion, and interaction with different species.

Therefore, we add the following sentence on Line 559-561.

“This indicates that the actual geographical range of an endangered tree species depends largely on its suitable habitat, as well as on its origin and evolution, on propagule dispersion, and interaction with different species”

As for the second suggestion, we add the following sentence on Line 563-565, and make a modification.

Please see Line 559-561, 563-565.

  1. "Section 4.5

Propose clear conservation strategies and recommend specific actions such as:

In Situ conservation: Expand existing nature reserves in highly suitable areas.

Ex Situ conservation: Develop seed banks and botanical gardens to maintain genetic diversity.

Restoration: Reconnect fragmented habitats through ecological corridors."

√ Thank you very much for your comment. We have accepted the suggestion.

Please see Line 642-648.

Round 2

Reviewer 1 Report

Comments and Suggestions for Authors

The introduction is still too long, and the knowledge gap is unclear.

Also, my concerns raised last time were not fully fixed.

Comments on the Quality of English Language

need to be concise

Reviewer 2 Report

Comments and Suggestions for Authors

Dear Authors,

I have carefully reviewed your revised manuscript and am pleased to note that you have responded thoroughly to my earlier suggestions. The changes you have made have improved the quality and clarity of the work, and I believe it is ready for publication in Biology.
